# Bias-inducing geometries: an exactly solvable data model with fairness implications

## Abstract

Machine learning (ML) may be oblivious to human bias but it is not immune to its perpetuation. Marginalisation and iniquitous group representation are often traceable in the very data used for training, and may be reflected or even enhanced by the learning models. In the present work, we aim to clarify the role played by data geometry in the emergence of ML bias. We introduce an exactly solvable high-dimensional model of data imbalance, where parametric control over the many bias-inducing factors allows for an extensive exploration of the bias inheritance mechanism. Through the tools of statistical physics, we analytically characterise the typical properties of learning models trained in this synthetic framework and obtain exact predictions for the observables that are commonly employed for fairness assessment. Simplifying the nature of the problem to its minimal components, we can retrace and unpack typical unfairness behaviour observed on real-world datasets. Finally, we focus on the effectiveness of bias mitigation strategies, first by considering a loss-reweighing scheme, that allows for an implicit minimisation of different unfairness metrics and a quantification of the incompatibilities between existing fairness criteria. Then, we propose a mitigation strategy based on a matched inference setting that entails the introduction of coupled learning models. Our theoretical analysis of this approach shows that the coupled strategy can strike superior fairness-accuracy trade-offs.

## Introduction

Machine Learning (ML) systems are actively being integrated into multiple aspects of our lives, making the question about their failure points of utmost importance. Recent studies (Buolamwini & Gebru, 2018; Weidinger et al., 2021) have shown that these systems may have a significant disparity in failure rates across the multiple sub-populations targeted in the application. ML systems appear to perpetuate discriminatory biases that align with those present in our society (Benjamin, 2019; Noble, 2018; Eubanks, 2018; Broussard, 2018).

Bias could originate at many levels in the ML pipeline, from the problem definition to data collection, to the training and deployment of the ML algorithm (Suresh & Guttag, 2021). Without minimising the importance of the other factors, we will focus this study on data itself, which often represents a critical source of bias (Perez, 2019). A dataset can inadvertently contain the record of a history of discriminatory behaviour, tangled in complex dependencies which are hardly eradicated even when the explicit discriminatory attribute is removed. The root of the discrimination can indeed be hidden in the structural properties of the dataset, since different sub-populations are almost inevitably heterogeneously represented. Thus, an important open question is *when and how such heterogeneity can induce bias in ML systems.*

Disproportional numerical representation of the different sub-populations in a dataset is of course the most visible – but not only possible – form of representation heterogeneity. Learning with an unbalanced dataset, where some classes are underrepresented, has been shown to drastically bias the outcome of a classifier (Kotsiantis et al., 2006; Wang et al., 2021). Furthermore, imbalances in the relative representation can become particularly problematic in the high-dimensional, feature-rich regime (Chen & Wasikowski, 2008). In this work, however, we aim at identifying the *many other geometrical properties of data that can systematically lead to biased trained models.*

ML bias can be prevented or removed by implementing targeted heuristics in the training pipeline. A vast literature focuses on the study of bias mitigation methods in the context of real-world data, either by revising the data sampling step or by adjusting the optimisation objective. Several methods have been shown to be effective in correcting for class imbalances in standard classification settings, including oversampling (Chawla et al., 2002), undersampling (Liu et al., 2008) and reweighing strategies. In the general framework, the class

label and sub-population membership do not necessarily overlap but some of these ideas can be adapted to allow for bias mitigation (Wang et al., 2020; Idrissi et al., 2022). Despite many empirical successes, a large gap remains in the theoretical understanding of bias-induction mechanisms and how to counteract them. The introduction of a *controlled minimal setting*, where these phenomena can be characterised exactly, could allow for a better theoretical grasp of these nuanced interactions.

In this work, we aim to address this theory gap by introducing the *Teacher-Mixture* (T-M) model, a novel exactly-solvable generative model producing high-dimensional correlated data. This model offers a controlled setting where data imbalances and the emergence of bias become more transparent and can be better understood, allowing also for the design of theoretically grounded and effective solutions. The model is designed to capture common observations about the data structure of real datasets, with a particular focus on the coexistence of non-trivial correlations, both among inputs and between inputs and labels, induced by the presence of a sub-population structure. Surprisingly, the few ingredients encoded in the model are capable of generating a rich and realistic ML bias phenomenology.

The rest of the work is structured as follows: in Sec. 1 we describe the T-M model and derive an analytical characterisation of the typical performance of solutions in the high-dimensional limit. Sec. 2 examines the different sources of bias (shown in Fig. 1B) and their role in the bias-induction mechanism in a sub-population-agnostic shallow network. This leads also to the identification of a *positive transfer* effect among the sub-populations within the dataset: despite their distinct characteristics, which make it tempting to split the dataset and use different classifiers, the shared underlying features can be leveraged to enhance the performance of a single classifier on both groups. Finally in Sec. 3, we focus on the problem of mitigating bias when the membership information is accessible. We theoretically analyse the effects of a sample reweighing mitigation strategy, highlighting the trade-offs between different definitions of fairness. We also propose and analyse a model-matched mitigation strategy, where two coupled networks are jointly trained, allowing for specialisation on different sub-populations as well as transfer of valuable cross-population information.

# 1 Modelling Data Imbalance

Drug testing provides a historically significant example of the potential consequences of unchecked data imbalance: substantial evidence (Hughes, 2007; 2019; Perez, 2019) shows that the scarcity of data points corresponding to female individuals in drug-efficiency studies resulted in a larger number of side effects in their group. This historical data gap has often been justified on the basis of a "simplification" criterion: due to the inherent variance of the female sub-population (caused e.g. by fluctuating hormonal levels), their inclusion in medical trials can introduce complex interactions that are instead absent in the "standard" male sub-population. However, ignoring biological sex as a discriminative factor in the analysis can induce serious adverse effects on the female sub-population, ranging from over-dosage to ineffectiveness of treatment.

The Teacher-Mixture (T-M) model is designed to allow a theoretical characterisation of the impact of such data imbalances on the inference process (e.g., determining a discriminative rule for administering the drug to the patients). While retaining analytical tractability, the T-M model retraces the main features of real data with multiple coexisting sub-populations and allows for a richer phenomenology than previously analysed data models. In Fig. 1A, we sketch a 2-dimensional cartoon of the T-M data distribution, framed in the context of drug testing.

The T-M combines aspects of two common modelling frameworks for supervised learning, namely the Gaussian-Mixture (GM) and the Teacher-Student (TS) setups (Zdeborová & Krzakala, 2016). The GM is a simple model of clustered input data, where each data point is sampled from one out of a narrow set of high-dimensional Gaussian distributions. Instead, the T-M inherits from the TS setup a simple model of input-label correlation, where the ground-truth labels are produced by a realisable "teacher" rule, to be inferred by the trained model during the learning process. For simplicity, in this study, we will only consider linear labelling rules. In the T-M, however, we allow for the existence of group-specific rules: at inference, the model will have to strike a compromise between them. Many different factors, parametrically controlled in the T-M, can generate bias in a classifier, T-M allows to explore different realisation of the problem as shown in Fig. 1B.

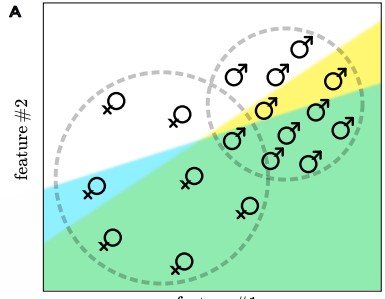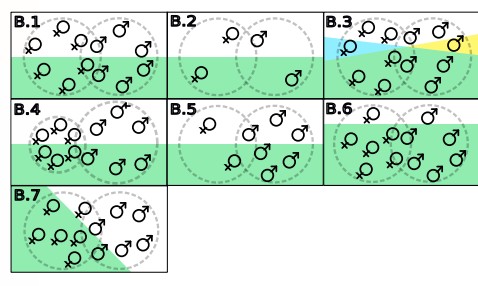

Figure 1: **The Teacher-Mixture (T-M) model can account for several types of data imbalance**.
**Panel A** The T-M model is a generative model of high-dimensional structured data. Inputs are sampled
from a combination of multivariate Gaussian distributions, with different centroids and covariances for each
sub-population in the dataset. The probability of sampling from each sub-population can be tuned, giving
rise to representation imbalance. In particular, the cartoon shows a larger relative representation for the
male population ($\sigma$), which also has a smaller variance. The cyan and yellow shaded regions (green in their
intersection) denote the decision boundaries of the labelling rules for the different data sub-populations, which
in principle can be misaligned. **Panel B** The panel exemplifies how manipulating the parameters of the
T-M model can alter the data distribution: B.1 represents the *balanced condition* with equally represented,
distributed and labelled samples; B.2 shows *scarcity* of data points in both clusters; B.3 displays an example of
*rule misalignment*; B.4 shows different sub-population *variances*; B.5 shows *relative representation* imbalance;
B.6 represents the case of *unbalanced labels*; B.7 shows a case of positive *group-label correlation* .

In the sketch in Fig. 1A, the female and male sub-populations are represented as partially overlapping data
clouds, with different variances and group-dependent offsets (the two features in the sketch could represent
some combination of clinical values recorded during the trial). Note that the female population is numerically
under-represented, as in the above-described real-world scenarios. The shaded areas represent the true regions
of the effectiveness of the tested drug for female/male subjects (cyan/yellow shades). As depicted in Fig. 2A,
the goal of the inference model is to infer a decision boundary for the administration of the drug based on
the observations of its effectiveness on the test subjects. While the vast majority of the subjects would be
identically classified according to the two different labelling rules (green region), some false positives could
occur if the inference only accounts for a single sub-population.

Fig. 2 shows a representation of student trained on the T-M model. If no explicit bias mitigation strategy is
employed, a heterogeneous representation of the two sub-groups will inevitably lead to a biased classifier.
In panels B and C of Fig. 2, we show that the classification accuracy on the two sub-populations, as the
fraction of data points belonging to each group (the relative representation) is varied, is biased in favour of
the majority group.

For simplicity, the results discussed in this paper will focus on the case of two groups, but the analysis could
be extended to multiple sub-populations.

**Formal definition** We consider a synthetic dataset of $n$ samples $\mathcal{D} = \{\mathbf{x}^\mu, y^\mu\}_{\mu=1}^n$, with $\mathbf{x}^\mu \in \mathbb{R}^d$, $y^\mu \in \{1, -1\}$. We define the $\mathcal{O}(1)$ ratio $\alpha = n/d$ and we refer to it as the dataset size parameter. Each
input vector is i.i.d. sampled from a mixture of two symmetric Gaussians with variances $\Delta = \{\Delta_+, \Delta_-\}$,
$\mathbf{x} \sim \mathcal{N}(\pm\boldsymbol{v}/\sqrt{d}, \Delta_\pm \mathbb{I}^{d \times d})$, with respective probabilities $\rho$ and $(1 - \rho)$. The shift vector $\mathbf{v}$ is a Gaussian vector
with i.i.d. entries with zero mean and variance 1. The $1/\sqrt{d}$ scaling corresponds to the *high-noise* noise
regime, where the two Gaussian clouds are overlapping and hard to disentangle (Mignacco et al., 2020;
Saglietti & Zdeborová, 2022), e.g. as in the case of CelebA and MEPS shown in the Appendix C. The
ground-truth labels, instead, are provided by two Gaussian teacher vectors, namely $\mathbf{W}_T^+$ and $\mathbf{W}_T^-$, with
respective bias terms $b_T^+$ and $b_T^-$, normalized to the $d$-dimensional sphere $\frac{1}{d}\mathbb{E}[\|\mathbf{W}_T^\pm\|^2] = 1$ and with mutual
overlap $\frac{1}{d}\mathbb{E}[\mathbf{W}_T^+ \cdot \mathbf{W}_T^-] = q_T$. Each teacher produces labels for the inputs with the corresponding group-
membership, namely $y^\mu = \text{sign}\left(\mathbf{W}_T^{c^\mu} \cdot \mathbf{x}^\mu/\sqrt{d} + b_T^{c^\mu}\right)$, with $c^\mu \in \{+, -\}$. The teacher bias terms are included
in the model to control the fraction of positive and negative samples within the two sub-populations. Overall,
the geometric picture of the data distribution (sketched in Fig. 1A) is summarised by three sufficient statics,

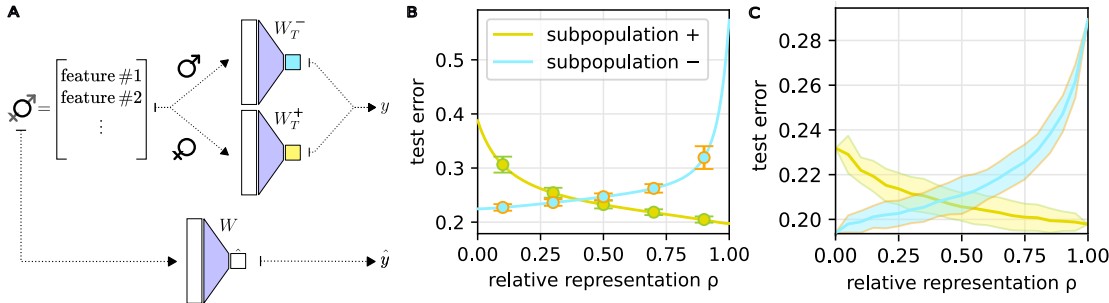

Figure 2: **Training on T-M model and comparison between error on synthetic and real data**. **Panel A** Given a vector of input features and a group membership (male/female), the ground-truth label is assigned by the associated 1-layer teacher network (represented by one of the vectors $W_T^{\pm}$). The decision boundaries are demarked in blue and yellow (while their intersection is coloured in green)). The labelling rules can be aligned, i.e. the decision rule does not depend on the group membership, or misaligned as in panel A. A 1-layer student network is given inputs $\boldsymbol{x}^{\mu}$ and labels $y^{\mu}$, and trained to produce the correct outputs $\hat{y}$ via gradient descent on the loss $\ell(\hat{y}, y)$. **Panel B** shows the test performance (on the two sub-populations) for a student network trained on mixed data instances with variable relative representations. Unsurprisingly, when one sub-population is largely predominant in the dataset, the classifier becomes biased to have higher accuracy on it. The plot shows the match between the analytic curves described in Sec. 1 (solid lines), and numerical simulations on the synthetic framework (dots). **Panel C** contains a similar experiment, but with data from the 'CelebA' dataset (Liu et al., 2015). Details in the Appendix C.

$m_T^{\pm} = \frac{1}{d} \boldsymbol{W}_T^{\pm} \cdot \boldsymbol{v}$ and $q_T$, that respectively quantify the alignment of the teacher labelling rules with respect to the shift vector, controlling the group-label correlation, and the alignment between the teacher vectors, controlling the correlation between labels assigned to similar inputs belonging to different communities.

Given the synthetic dataset $\mathcal{D}$, we study the properties of a single-layer network $\boldsymbol{W}$, with bias term $b$, producing outputs $\hat{y}^{\mu} = \text{sign}\left(\boldsymbol{W} \cdot \mathbf{x}^{\mu}/\sqrt{d} + b\right)$, and trained via empirical risk minimisation (ERM) with loss:

$$\mathcal{L}(\boldsymbol{W}, b) = \sum_{\mu \in \mathcal{D}} \ell\left(\boldsymbol{W}, b; \boldsymbol{x}^{\mu}, y^{\mu}\right) + \frac{\lambda ||\boldsymbol{W}||_2^2}{2} \tag{1}$$

where $\ell$ is assumed to be convex in student's parameters and $\lambda$ is an external parameter that regulates the intensity of the $L_2$ regularisation.

Given this framework, we derive a theoretical characterisation of the training performance of this learning model and consider the possible implications from an ML fairness perspective. In particular, we aim to study the role of data geometry and cardinality in the training of a fair classifier. To quantify the level of bias in the predictions of the trained model, we need to choose a metric of fairness. We will employ *disparate impact* (DI) (Feldman et al., 2015), an ML analog of the 80% rule (Commission et al., 1979), which allows a simple assessment of the over-specialisation of the classifier on one of the sub-populations. In our framework, we characterise bias against sub-population + using the following definition of

$$DI = \frac{p(\hat{y} = y|+)}{p(\hat{y} = y|-)}, \tag{2}$$

evaluating the ratio between test accuracy in sub-population + and sub-population −. Note that how to measure bias is itself an active line of research, and the DI alone cannot return a full picture of the unfairness. In Sec. 3, we compare these results with those obtained with other metrics. Notice, that the T-M model allows to parametrically move from a model-mismatched scenario ($q_T < 1$) where the rule to be inferred is not in the function space of learnable rules, to a model-matched scenario ($q_T = 1$) where the rule is actually learnable but, as we will discuss further in Sec. 2, the model may systematically fail to identify it. We will discuss in detail when these failure modes occur and why.

Finally, the T-M model has, at the same time, the advantage of being simple, allowing a better understanding of the many facets of ML bias, and the disadvantage of being simple, since some modelling assumptions might not reflect the complexity of real-world data. For example, we ignore any type of correlation among

the inputs other than the clustering structure. However, this modelling approach continues a long tradition of research in statistical physics (Charbonneau et al., 2023), which has shown that theoretical insights gained in prototypical settings can often be helpful in disentangling and interpreting the complexity of real-world behaviour.

**Remark 1** *By looking at the available degrees of freedom in the T-M, several possible sources of bias naturally emerge from the model:*

- *the* relative representation, $\rho = n_+/(n_+ + n_-)$, *with* $n_c$ *the number of points in group* $c$, $c \in \{+, -\}$.

- *the* group variance, $\Delta_c$, *determining the width of the clusters.*

- *the* group label frequencies, *controlled through the bias terms* $b_T^c$.

- *the* group-label correlation, $m_T^c$.

- *the* inter-group similarity, $q_T$, *which measures the alignment between the two teachers, i.e. the linear discriminators that assign the labels to the two groups of inputs.*

- *the* dataset size, $\alpha$, *representing the ratio between the number of inputs and the input dimension.*

**Theoretical analysis in high-dimensions.** In principle, solving Eq. 1 requires finding the minimiser of a complex non-linear, high-dimensional, quenched random function. However, statistical physics (Mézard et al., 1987) showed that in the limit $n, d \to \infty$, $n/d = \alpha$, a large class of problems, including the T-M model, becomes analytically tractable. In fact, in this proportional high-dimensional regime, the behaviour of the learning model becomes deterministic and trackable due to the strong concentration properties of a narrow set of descriptors that specify the relevant geometrical properties of the ERM estimator. The original high-dimensional learning problem can be reduced to a simple system of equations that depends on a set of scalar sufficient statistics, $\Theta = \{Q = \frac{1}{d}\boldsymbol{W} \cdot \boldsymbol{W}, m = \frac{1}{d}\boldsymbol{W} \cdot \boldsymbol{v}, R_\pm = \frac{1}{d}\boldsymbol{W} \cdot \boldsymbol{W}_T^\pm, \delta q, b\}$, respectively representing the typical norm of the trained estimator, its magnetisation in the direction of the cluster centres, its alignment with the two teacher vectors of the T-M, the rescaled variance of the self-overlap (details in Appendix B), and the student bias term.

The results below summarise the main findings of the replica analysis, while all the technical details are reported in Appendix B.

**Analytical result 1** *Given a specific setup of the T-M model (formal definition of the model given above, model parameters recapped in Appendix A), in the high dimensional limit when $n, d \to \infty$ at a fixed ratio $\alpha = n/d$, the scalar descriptors $\Theta = \{Q, m, R_\pm, \delta q, b\}$ of the vector $\boldsymbol{W}$ obtained by empirical risk minimisation Eq. 1 with a generic convex loss $\ell$, and their Lagrange multipliers $\hat{\Theta} = \{\hat{Q}, \hat{m}, \hat{R}_\pm, \delta\hat{q}\}$, converge to deterministic quantities given by the unique fixed point of the system:* $Q = -2\frac{\partial s(\hat{\Theta}; \lambda)}{\partial \delta\hat{q}}$; $m = \frac{\partial s(\hat{\Theta}; \lambda)}{\partial \hat{m}}$; $R_\pm = \frac{\partial s(\hat{\Theta}; \lambda)}{\partial \hat{R}_\pm}$; $\delta q = 2\frac{\partial s(\hat{\Theta}; \lambda)}{\partial \hat{Q}}$; $\hat{Q} = 2\alpha\frac{\partial e(\Theta; \Delta)}{\partial \delta q}$; $\hat{m} = \alpha\frac{\partial e(\Theta; \Delta)}{\partial m}$; $\hat{R}_\pm = \alpha\frac{\partial e(\Theta; \Delta_\pm)}{\partial R_\pm}$; $\delta\hat{q} = 2\alpha\frac{\partial e(\Theta; \Delta)}{\partial Q}$. *with:*

$$s(\hat{\Theta}; \lambda) = \frac{1}{2(\delta\hat{q} + \lambda)}\left[\hat{Q} + \left(\hat{m} + \sum_{c \in \{\pm\}} m_T^c \hat{R}_c\right)^2 + \sum_{c \in \{\pm\}}\left(1 - (m_T^c)^2\right)\hat{R}_c^2 + 2\left(q_T - \prod_{c \in \{\pm\}} m_T^c\right)\prod_{c \in \{\pm\}}\hat{R}_c\right] \tag{3}$$

$$e(\Theta; \Delta) = \mathbb{E}_c\left[\mathbb{E}_z \sum_{y=\pm 1} H\left(-y\frac{\sqrt{Q}(c\, m_T^c + b_T^c) + \sqrt{\Delta_c}R_c z}{\sqrt{\Delta_c(Q - R_c^2)}}\right)v(y, c, \Theta)\right] \tag{4}$$

*where* $c \in \{+, -\} \sim \text{Bernoulli}(\rho)$, $z \sim \mathcal{N}(0, 1)$, $H(\cdot) = \frac{1}{2}\text{erfc}(\cdot/\sqrt{2})$ *is the Gaussian tail function. The function* $v(y, c, \Theta)$, *appearing in Eq. 4, depends parametrically on the scalar descriptors and entails a 1-dimensional optimization problem:* $v(y, c, \Theta) = \max_w\left[-\frac{w^2}{2} - \ell\left(y, \sqrt{\Delta_c\delta q}w + \sqrt{\Delta_c Q}z + c\, m + b\right)\right]$. *The student bias term $b$ implicitly solves the equation* $\partial_b e(\Theta; \Delta) = 0$. *Eqs. 3 and 4 represent the so-called entropic and energetic contributions appearing in the quenched free-entropy of the system (details in Appendix B).*

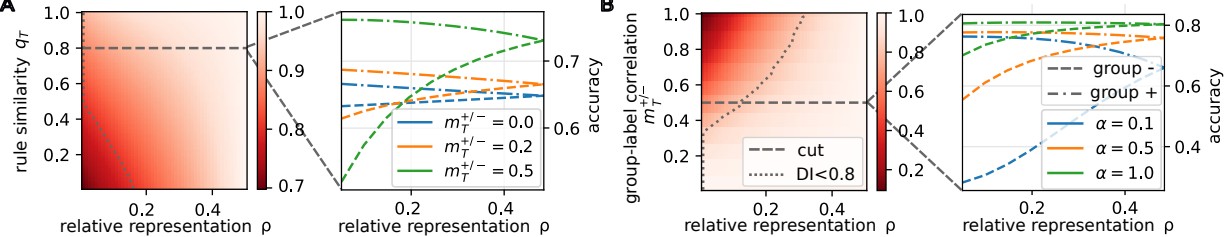

Figure 3: **Simple geometrical properties cause the emergence of bias.** Each point in the left diagrams shows, for different values of the model parameters, the Disparate Impact (DI) of the trained model (darker colours represent stronger biases). In particular, in the left diagrams, on the x-axis we vary the relative representation $\rho$, while on the y-axis we explore possible values of the rule similarity $q_T$ for **Panel A** and the group-label correlation $m_T^\pm$ for **Panel B**. The corresponding figures on the right show the values of the accuracy for the two sub-populations in correspondence of the cut represented by the dashed line on the left.

The yielded fixed point values for the scalar descriptors, $\Theta$, can be used to obtain deterministic predictions for common model evaluation metrics, such as the *confusion matrix* or the *generalisation error*, in high-dimensional realizations of the system.

The presented result was obtained through the non-rigorous replica method from statistical physics (Mézard et al., 1987; Engel & Van den Broeck, 2001; Zdeborová & Krzakala, 2016). The derivation details are deferred to the Appendix B. We remark that, in convex settings, the replica method was rigorously proven to yield exact results in a range of different model settings. In particular, a lengthy but straightforward generalization of the proofs presented in (Thrampoulidis et al., 2015; Mignacco et al., 2020; Loureiro et al., 2021) could be derived for the T-M case, but this is out of the scope of the present work. In this manuscript, we verify the validity of our replica theory by comparison with numerical simulations, as shown e.g. in Fig. 2B.

**Analytical result 2** *In the same limit as in Analytical result 1, the entries of the confusion matrix, representing the probability of classifying as $\hat{y}$ an instance sampled from sub-population $c$ with true label $y$, are given by:*

$$p\left(\hat{y} \mid y; c\right) = \mathbb{E}_z\left[\text{Heav}\left(y\left(\sqrt{\Delta_c}z + c\, m_T^c + b_T^c\right)\right) H\left(-\hat{y}\frac{(c\, m + b) + \sqrt{\Delta_c}R_c z}{\sqrt{\Delta_c(Q - R_c^2)}}\right)\right], \tag{5}$$

*where $z \sim \mathcal{N}(0, 1)$ and $\text{Heav}(\cdot)$ is the Heaviside step function. The generalisation error, representing the fraction of wrongly labelled instances, can then be obtained as $\epsilon_g = \mathbb{E}_c\left[\sum_{\hat{y}\neq y} p(\hat{y} \mid y; c)\right]$.*

This second result yields a fully deterministic estimate of the accuracy of the trained model on the different data sub-populations. These scores will be used in the following sections to investigate the possible presence of bias in the classification output of the model. In particular, they will be useful to estimate numerator and denominator of the *DI*, Eq. 2. Note that the results 1 and 2 allow for an extremely efficient and exact evaluation of the learning performance in the T-M, remapping the original high-dimensional optimisation problem onto a system of deterministic scalar equations that can be easily solved by recursion.

## 2 Investigating the sources of bias

With these analytical results in hand, we now turn to systematically investigating the effect of the sources of bias identified in remark 1, which potentially mine the design of a fair classifier. We specialise on cross-entropy loss and perform three separate experiments to summarise some distinctive features of the fairness behaviour in the T-M: namely, the impact of the correlation between the labelling rules and the group structure, the interplay between relative representation and group variance, and the different accuracy trade-offs between the sub-populations at different dataset sizes. The parameters of the experiments, if not specified in the caption, are detailed in the Appendix B.1.

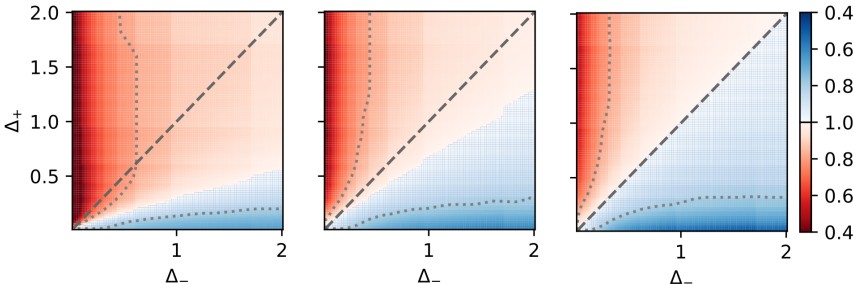

Figure 4: **Emergence of bias even in balanced datasets.** We show the disparate impact as the distribution of the two subpopulations is changed by altering their variances ($\Delta_+$ and $\Delta_-$). The diagonal line gives the configurations where the two subpopulations have the same variance. The two figures consider different levels of representation, from left to right $\rho = 0.1, 0.3, 0.5$. The latter is the situation with both subpopulations being equally represented in the dataset. We use the red and blue colours to quantify the disparate bias against sub-population $+$ and $-$ (respectively).

## 2.1 Group-label correlation.

In Fig. 3A, we consider a scenario where the labelling rules for the two groups are not perfectly aligned, i.e. $\boldsymbol{W}_T^+ \neq \boldsymbol{W}_T^-$ (and/or $b_T^+ \neq b_T^-$). Note that, in this case, we have a clear mismatch between the learning model, a single linear classifier, and the true input-output structure in the data: the learning model cannot reach perfect generalisation for both sub-populations at the same time. For simplicity, we set an equal correlation between the two teacher vectors and the shift vector, $m_T^+ = m_T^- > 0$, and isolate the role of rule similarity $q_T$. The upper-left panel shows a phase diagram of the DI (DI$< 1$ indicating a lower accuracy on group $+$), as function of the similarity of the teachers and the fraction of $+$ samples in the dataset. As intuitively expected, the induced bias exceeds the 80% rule when the labelling rules are misaligned and the group sizes are numerically unbalanced (small $q_T$ and $\rho$). Indeed, in the cut displayed in the upper-right panel, by lowering the group-label correlation $m_T^\pm$ the gap between the measured accuracies on the two sub-populations becomes smaller. However:

**Remark 2** *Even when $q_T = 1$ and the task is solvable (i.e. the classifier can learn the input-output mapping), the trained model can still be biased.*

This is shown in Fig. 3B, where a large high-bias region (DI$< 80\%$) exists. In particular, the lower-left panel shows the cause of this effect in the presence of a non-zero group-label correlation $m_T^\pm$, and in the lower-right panel we see how this effect is more pronounced in the data-scarce regime. In all four panels, as $\rho$ reaches 0.5, the two sub-populations become equally represented and the classifier achieves the same accuracy for both.

## 2.2 Bias and variance.

In Fig. 4, we plot the DI as a function of the group variances $\Delta_\pm$, for different values of the fraction of $+$ samples. One finds that the model might need a disproportionate number of samples in the two groups to obtain comparable accuracies. We can see that:

**Remark 3** *Balancing the group relative representation does not guarantee a fair training outcome.*

In fact, the quality of a group's representation in the dataset can increase if the number of points is kept constant but the group variance is reduced. The blue regions in the left panel indicate a higher accuracy for the smaller sub-population even if the dataset only contains 10% of samples belonging to it. This exemplifies the fact that a very focused distribution (low $\Delta_\pm$) actually requires less samples. The right panel ($\rho = 0.5$) shows the scenario one would expect *a priori*: on the diagonal line the DI is balanced, but by setting $\Delta_+ > \Delta_-$ (or viceversa) one induces a bias in the classification.

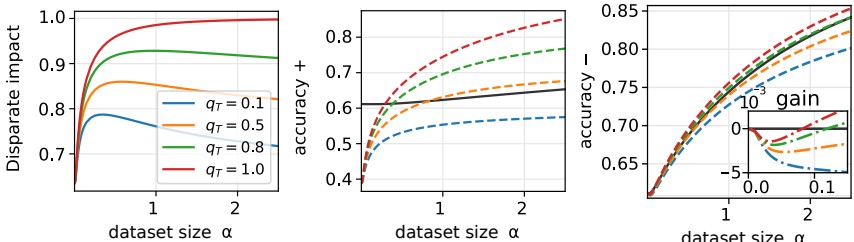

Figure 5: **Performance benefits for both subpopulations under shared training.** With 10% of the data points in sub-population $+$ ($\rho = 0.1$), we compare the performance with different levels of rule similarity ($q_T$) as the size of the dataset is increased, showing the disparate impact in the left figure and the individual accuracies in central and right ones. In central and right figures, the baselines –plotted in black– show the accuracies attained when the model is trained only on the corresponding group data. The inset of the rightmost figure highlights the differences in accuracy in the small dataset regime. When the rules are sufficiently aligned, joint training on both groups will induce a better accuracy on the smaller sub-population provided $\alpha$ is not too small. Moreover, at intermediate values of $\alpha$ also the larger group can benefit from the information transfer.

### 2.3 Positive transfer.

If mixing different sub-populations in the same dataset can induce unfair behaviour, one could think of splitting the data and train independent models. In Fig. 5, we show that a *positive transfer effect* (Gerace et al., 2022) can yet be traced between the two groups when the rules are sufficiently similar. This means that the accuracy on the under-represented group is enhanced when information is shared across the two sub-populations.

**Remark 4** *The performance on the smaller sub-population tends to further deteriorate if the dataset is split according to the sub-group structure.*

To clarify this point, in the left plot of Fig. 5 we show the DI as a function of the dataset size $\alpha$, for several values of the rule similarity $q_T$ and at fixed $\rho = 0.10$. In the centre and right plots in Fig. 5, we also display the gain in accuracy on each sub-population when the model is trained on the full dataset, comparing with a baseline classifier (black lines) trained only on the respective data subsets ($+$ in the central panel, $-$ in the lower panel). These two plots elucidate the positive transfer effect: for sufficiently similar rules (large $q_T$), both populations can benefit from shared training at intermediate dataset sizes. If the dataset is too small (low $\alpha$ regime), the lack of data combined with a high variance in the input distribution can induce over-fitting, with a larger drop in performance for the smaller group. On the other hand, as the dataset size becomes sufficiently large, the positive transfer effect is eventually lost for the large sub-population (large $\alpha$ regime).

Connecting the results of this section to the drug testing examples, after observing that the distribution of side effects in the female population presented a larger variance, the solution was simply to collect more data of female subjects. Moreover, the results shown in Fig. 4 indicate the need for a higher relative representation of female subjects in the dataset to achieve an unbiased classifier. While implementing this solution might have introduced higher variance in the results due to the intrinsic high-variability of the data, it would have significantly reduced the risk of administering drugs with limited testing on half of the population.

## 3 Mitigation strategies

To assess or ensure the fairness of a ML model on a given data distribution, a plethora of different fairness criteria have been designed (Speicher et al., 2018; Castelnovo et al., 2022). In convex settings, any of these criteria can be separately enforced via a hard constraint during the optimisation process (Agarwal et al., 2018; 2019; Celis et al., 2019). However, it was proved that some criteria are completely incompatible and cannot be exactly achieved simultaneously (Kleinberg et al., 2016; Corbett-Davies & Goel, 2018; Barocas et al., 2019). In the same spirit of (Speicher et al., 2018), we drop the hard constraint and instead quantify exactly how far a given trained model is from meeting the criteria. Each criterion requires the probability of obtaining a specific classification outcome $E$ to be the same across the sub-populations. For example,

| FAIRNESS METRIC | CONDITION |
|---|---|
| *Statistical Parity* | $\mathbb{P}[\hat{Y} = y\|C = c] = \mathbb{P}[\hat{Y} = y] \; \forall y, c$ |
| *Equal Opportunity* | $\mathbb{P}[\hat{Y} = 1\|C = c, Y = 1] = \mathbb{P}[\hat{Y} = 1\|Y = 1] \; \forall c$ |
| *Equal Accuracy* | $\mathbb{P}[\hat{Y} = y\|C = c, Y = y] = \mathbb{P}[\hat{Y} = y\|Y = y] \; \forall y, c$ |
| *Equal Odds* | $\mathbb{P}[\hat{Y} = 1\|C = c, Y = 1] = \mathbb{P}[\hat{Y} = 1\|Y = 1] \; \cap$ 
 $\mathbb{P}[\hat{Y} = 1\|C = c, Y = -1] = \mathbb{P}[\hat{Y} = 1\|Y = -1] \; \forall c$ |
| *Predicted Parity* | $\mathbb{P}[Y = 1\|C = +, \hat{Y} = y] = \mathbb{P}[Y = 1\|C = -, \hat{Y} = y] = \mathbb{P}[Y = 1\|\hat{Y} = y] \; \forall y$ |

Table 1: **List of Fairness Metrics.** *Statistical Parity*: Equal fractions of each group should be treated as belonging to the positive class (Dwork et al., 2012; Kleinberg et al., 2016; Corbett-Davies et al., 2017). *Equal Opportunity*: Each group needs to achieve equal true positive rate (Hardt et al., 2016). *Equal Accuracy*: Each group is required to achieve the same level of accuracy. *Equal Odds*: Each group should achieve equal true positive and false positive rates(Feldman et al., 2015; Zafar et al., 2017). *Predicted Parity*. Given inputs that are classified by the model with label $y$, the fraction of input with true label $y^*$ should be consistent across sub-populations. This gives two sub-criteria: *predicted parity 1* requires the condition only for $y^* = 1$, while *predicted parity 10* requires the condition for both $y^* = 1$ and $y^* = -1$ (Chouldechova, 2017).

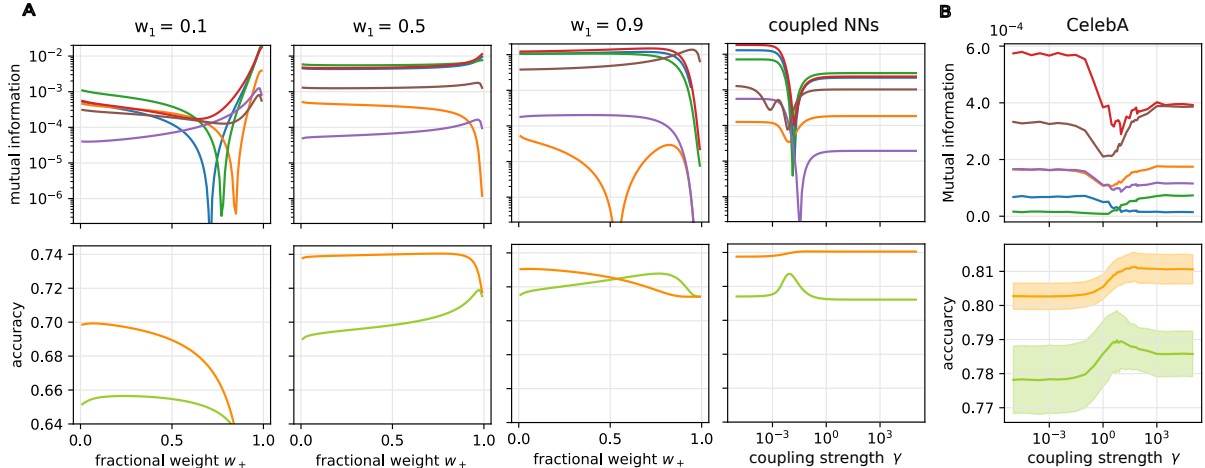

Figure 6: **Fairness-accuracy trade-off with reweighing and coupled architecture. Panel A** The figures show the effect of re-weighting and coupled architectures de-biasing methods in a instance of the T-M model. The lowers figures shows the accuracy for subpopulation $+$ and subpopulation $-$ and the upper figures show the mutual information for the several fairness metrics defined in Table 1, namely statistical parity, equal opportunities, equal accuracy, equal odds, predicted parity 1, and predicted parity 10. The goal of the algorithm is to identify regions with high accuracy (lower figures) and low mutual information (higher figures) for all metrics: this would imply that fairness is approximately achieved under all the criteria. The first three group of figures refer to the reweighing strategy, forcing higher relevance for a certain label in each panel ($w_1 = 0.1, 0.5, 0.9$) and the relative importance of a given subpopulation (parameter $w_+$) on the x-axis. The last panels instead refer to the proposed coupled networks strategy and the x-axis represent the strength of the coupling $\gamma$. The figures clearly show that our strategy achieves a higher accuracy in both subpopulations while preserving a higher level of fairness. Interestingly the minimum of the mutual information roughly correspond to the same parameter of the coupling strength, contrarily to what observed in the reweighing strategy. **Panel B** The two panels, show an example from the CelebA dataset splitting and classifying according to the attributes "Wearing_Lipstick" and "Wavy_Hair" respectively, more details are provided in the Appendix C.1 and C. The observations made for the synthetic model applies also in this real-world case.

according to the definition of *Equal Opportunity* (Table 1), the *true positive* rate $P(E = (\hat{Y} = 1\,|\,Y = 1))$ should not depend on the group-membership $C$. A natural measure of the observed dependence between $E$

and $C$ is given by the Mutual Information (MI):

$$I(E; C) = D_{KL}(\mathbb{P}[E, C] \mid \mathbb{P}[E]\mathbb{P}[C]) = \mathbb{E} \log \frac{\mathbb{P}[E, C]}{\mathbb{P}[E]\mathbb{P}[C]}. \tag{6}$$

The fairness condition is exactly verified only when the joint distribution factorizes, i.e. $\mathbb{P}[E, C] = \mathbb{P}[E]\mathbb{P}[C]$, and the mutual information goes to zero. Table 1 provides some other examples of classification events $E$, for some well-established fairness criteria. Note that some criteria might not be sensible in specific settings (e.g., *Statistical Parity* is unlikely to be guaranteed in a drug-testing scenario).

In the following, we consider two simple bias mitigation strategies that can be analysed within our analytical framework. The required generalisations of the results shown in Sec. 1 are detailed in the Appendix B. First, we study the de-biasing effect of a sample reweighing strategy where the relevance of each sample is varied based on its label and group membership (Kamiran & Calders, 2012; Plecko & Meinshausen, 2020; Lum & Johndrow, 2016). By adjusting the weights, one can indirectly minimise the MI relative to any given fairness measure. We use the simultaneous quantitative predictions on the various metrics to assess the compatibility between different fairness definitions. Then, we propose a theory-based mitigation protocol, along the lines of protocols used in the context of multi-task learning (Rusu et al., 2016), that couples two architectures trained in parallel.

**Loss Reweighing.** Recent literature shows that some fairness constraints cannot be satisfied simultaneously. ML systems are instead forced to accept trade-offs between them (Kleinberg et al., 2016). This sort of compromise is well-captured in the simple framework of the T-M model. The first three panels of Fig. 6a show accuracies and MI measured with respect to the various fairness criteria while varying the two reweighing parameters, $w_1$ and $w_+$, which up-weigh data points with true label 1 and in group +, respectively. Thus, each loss term in Eq. 1 is reweighed as:

$$\mathcal{W}(c, y) = \begin{cases} w_1 w_+ & \text{if} \quad c = +, y = 1 \\ w_1(1 - w_+) & \text{if} \quad c = -, y = 1 \\ (1 - w_1)w_+ & \text{if} \quad c = +, y = -1 \\ (1 - w_1)(1 - w_+) & \text{if} \quad c = -, y = -1. \end{cases} \tag{7}$$

By changing these relative weights one can force the model to pay more attention to some types of errors and re-establish a balance between the accuracies on the two sub-populations. The goal is to identify a classifier that achieves high accuracy (lower panels) while minimising the MI for different fairness metrics. Notably, given a weight $w_1$, these minima occur for different values of the weight $w_+$. Only $w_1 = 0.1$ seems to have a value of $w_+$ close to several minima of the MI, but this point correspond to a sharp decrease in accuracy in both subpopulations, thus fairness is achieved but at the expense of accuracy. These results are in agreement with rigorous results in the literature (Barocas et al., 2019), but also show how the incompatibilities between the different constraints extend to regimes where the fairness criteria are not exactly satisfied.

**Coupled Networks.** The emergence of classification bias in the T-M traces back to the clear mismatch between the generative model of data and the learning model. In order to move towards a matched inference setting, we need to enhance the learning model to account for the presence of multiple sub-populations and labelling rules. This inspires a mitigation strategy that we call *coupled neural networks*, consisting in the simultaneous training of multiple neural networks, each one seeing a different subset of the data associated with a different sub-population. This idea is represented by the following modified loss

$$\mathcal{L}_{\text{cnn}}(\boldsymbol{w}) = \sum_{c \in \pm} \sum_{\mu \in \mathcal{D}^c} \ell\left(\boldsymbol{W_c}, b_c; \boldsymbol{x}^\mu, y^\mu\right) + \frac{\lambda}{2}\left(||\boldsymbol{W}_+||_2^2 + ||\boldsymbol{W}_-||_2^2\right) + \frac{\gamma}{2}||\boldsymbol{W}_+ - \boldsymbol{W}_-||_2^2 \tag{8}$$

where $\boldsymbol{W}_\pm$ are the weights of the two networks, $b_\pm$ their associated bias terms, and $\hat{y}_\pm^\mu$ are their respective estimation of label $y^\mu$. The networks exchange information through the elastic penalty $\gamma$ that mutually attracts them, and the intensity of this elastic interaction is obtained by cross-validation. This approach shares some ideas with other methods present in the literature: (Zenke et al., 2017; Saglietti et al., 2021) add an elastic penalty term to the loss to bias the training trajectory, while (Calders & Verwer, 2010) proposes to combine estimations from different Bayesian classifiers. However, note that these prior approaches were tailored for different learning protocols or problem domains, and could not be applied in the problem setting considered in this paper. Other similar optimisation strategies include simultaneous linear regression (Myers & Myers, 1990) and multiple factor analysis, but to our knowledge, these approaches have not yet been applied in the bias mitigation context.

**Remark 5** *The coupled neural networks method allows for higher expressivity and specialisation on the various sub-populations, while also encouraging positive transfer between similarly labelled sub-populations, leading to better fairness-accuracy trade-offs.*

The upper rightmost plots of Fig. 6a, displaying the behaviour of the mutual information as a function of the coupling parameter for different fairness metrics, shows the key advantage of using this method. We observe a more robust consistency among the various fairness metrics: the positions of the different minima are now very close to each other. Moreover, the value of the coupling parameter achieving this agreement condition is also the one that minimises the gap in terms of test accuracy between the two sub-populations, as shown in the lower plot, without hindering the performance on the larger group. Notice that this result does not contradict the impossibility theorem (Barocas et al., 2019) which states that statistical parity, equal odds, and predicted parity cannot be satisfied altogether. In fact, our result only concerns soft minimisation of each fairness metrics. The result is in agreement with (Dutta et al., 2020) whose results show that the trade-off between fairness and accuracy vanishes when the true distribution of data is capture. Leveraging the universal approximation property of neural networks, the coupled networks method seems a promising direction for applications. In the panels of Fig. 6b we show promising preliminary results in the realistic dataset CelebA[1]. We stress that real data often presents more complex correlations than those modelled in the T-M, which may hinder the effectiveness of this strategy in unexpected ways.

The method of the coupled networks can be generalised to an arbitrary number of classes and sub-populations, and can be combined with standard clustering methods when the group membership label is not available. A future research direction will be to better investigate its range of applicability and, consequently, its limitations in real-world scenarios. In the Appendix B and B.1 we provide additional results for this method and we discuss the effect of training the networks on data subsets that only partially correlate with the true group structure.

## 4 Discussion

The goal of this study was to design a novel generative model of high-dimensional correlated data that allows the study of the effect of data geometry in the bias induction mechanism, in isolation from real-world confounding factors. While a focus on each specific dataset might be required to ensure fairness in applications with high societal impact, we believe that the study of the ML bias phenomenology in controlled synthetic settings might allow more coherent advancements in the understanding and prevention of bias induction.

The *Teacher-Mixture* (T-M) model captures non-trivial correlations among inputs and between inputs and labels, representing various imbalances appearing in real datasets when different sub-populations coexist in the sample. Surprisingly, with few modelling ingredients, the T-M can generate a rich and realistic ML bias phenomenology. We derive an analytical characterisation of its performance in the high-dimensional limit, showing agreement with numerical simulations and producing realistic unfairness behaviour. By isolating different sources of bias, we gain insights into situations where unfairness may persist despite apparent data balance, cautioning against relying solely on simple rebalancing techniques. We identify a positive transfer effect among diverse sub-populations, leveraging shared underlying features to enhance performance across groups. Additionally, we analysed the trade-offs between different ways of quantifying the model fairness, focusing on a sample reweighing mitigation strategy that can be analytically characterised within our framework. We also proposed a theory-based mitigation strategy that effectively promotes fairness without compromising overall performance, as demonstrated in the T-M model. Instead of imposing an hard constraint on a desired fairness metric which would incur in the incompatibility theorem (Barocas et al., 2019), the coupled networks strategy minimises several fairness metrics simultaneously only approximately. Furthermore, the strategy seems to avoid the typical fairness-accuracy trade-off. This result is in agreement with the findings of (Dutta et al., 2020) showing that it is possible to construct a Bayes optimal classifier that is not affected by the trade-off.

Moving forward, our model is extremely simplified with the respect to real data and practical architectures. Future directions for our research include incorporating more complex elements into the data model cosindering model complex data structures, for instance assuming that data live in a low-dimensional manifold of the input space (Goldt et al., 2020) or moving away from Gaussian setting (Adomaityte et al., 2024), and introducing the effect of feature dependencies (e.g., proxy variables) in the generated data. Another important but challenging address for further work is to move to the non-convex optimisation setting and to more

---

[1]The illustrated checkpoints are used only to show the similarity of behaviour in synthetic data and realistic data (CelebA), and not used or recommended to use in any face recognition systems or scenarios.

complex model architectures, where at this time some of the analytic techniques employed in this work fail to generalise. Howver, recent works succeeded in addressing some of this limitations, in particular considering the multi-label classification problem (Cornacchia et al., 2023) and more than a single layer (Loureiro et al., 2021)—despite still limited to random projection—these results could be included in future iterations of the work. Moreover, further explorations of the efficacy and limitations of the coupled networks strategy in the context of deep networks and more complex datasets is called for. By investigating its performance in deeper architectures and diverse real-world datasets, and connecting to the existing literature (Sagawa et al., 2020; Bell & Sagun, 2023), we can assess the scalability and generalisability of this approach for addressing fairness concerns.

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

# A  Symbols and notation

| | |
|---|---|
| $\mathbf{x}^\mu$ | $\mu$th data point in the training set |
| $y^\mu$ | $\mu$th label in the training set |
| $d$ | Input dimension |
| $n$ | Number of data points |
| $n_+$ | Number of data points from sub-population $+$ |
| $n_-$ | Number of data points from sub-population $-$ |
| $\alpha$ | Ration between number of data points and input dimension $n/d$ |
| $\ell$ | Loss (in particular cross-entropy loss used in the analysis) |
| $\lambda$ | Ridge-regulariser's strength |
| $\rho$ | Relative representation of sub-population $+$, $\rho = n_+/n$ |
| $\mathbf{W}_T^+$ | Teacher vector associated with sub-population $+$ |
| $\mathbf{W}_T^-$ | Teacher vector associated with sub-population $-$ |
| $\mathbf{W}_T^\pm$ | Shortcut notation to indicate both teacher vectors |
| $\mathbf{v}$ | Shift vector separating the two sub-populations |
| $\Delta_+$ | Variance of the sub-population $+$ |
| $\Delta_-$ | Variance of the sub-population $-$ |
| $b_T^+$ | Threshold (or bias term) associated with teacher $+$ |
| $b_T^-$ | Threshold associated with teacher $-$ |
| $b_T^\pm$ | Shortcut notation to indicate thresholds associated with both teachers |
| $q_T$ | Teacher-teacher overlap $q_T = \frac{1}{d}\mathbf{W}_T^+ \cdot \mathbf{W}_T^-$ |
| $m_T^+$ | Overlap between teacher $+$ and shift vector $m_T^+ = \frac{1}{d}\mathbf{W}_T^+ \cdot \mathbf{v}$ (group-label correlation) |
| $m_T^-$ | Overlap between teacher $-$ and shift vector $m_T^- = \frac{1}{d}\mathbf{W}_T^- \cdot \mathbf{v}$ |
| $m_T^\pm$ | Shortcut notation to indicate both overlaps between teachers and shift vector |
| $\boldsymbol{W}$ | Student weight vector |
| $\boldsymbol{W}_s$ | Weight vectors for the coupled student networks |
| $b$ | Student bias term |
| $b_s$ | Bias terms for the coupled student networks |
| $Q$ | Student-student overlap, $Q = \frac{1}{d}\mathbf{W} \cdot \mathbf{W}$ |
| $m$ | Overlap between student and shift vector, $Q = \frac{1}{d}\mathbf{W} \cdot \mathbf{v}$ |
| $R_+$ | Overlap between student and teacher associated with sub-population $+$, $Q = \frac{1}{d}\mathbf{W} \cdot \mathbf{W}_T^+$ |
| $R_-$ | Overlap between student and teacher associated with sub-population $+$, $Q = \frac{1}{d}\mathbf{W} \cdot \mathbf{W}_T^-$ |
| $R_\pm$ | Shortcut notation to indicate overlap between student and both teachers |
| $\delta q$ | Variance of the self-overlap at finite temperature (see Appendix B) |
| $\omega_+$ | In the re-weighing strategy, indicates the coefficient associated with data from sub-population $+$ |
| $\omega_1$ | In the re-weighing strategy, indicates the coefficient associated with data labelled 1 |
| $\gamma$ | In the coupled networks strategy, indicates the coupling coefficient |

# B  Replica analysis

We will directly present the most general setting for this calculation, where the learning model is composed of two linear classifiers ("students" in the following), coupled by an elastic penalty of intensity $\gamma$. This allows us to characterise the novel mitigation strategy proposed in this work, while the standard case with a single learning model can be obtained by setting $\gamma = 0$. Each student, denoted by the index $s = 1, 2$, is assumed to

be trained on a fraction of the full dataset $\mathcal{D}_s$. Note that, in principle, the data split could not be aligned with the group structure of the dataset.

The loss function for the coupled learning model reads:

$$\mathcal{L}(\boldsymbol{W}_1, \boldsymbol{W}_2) = \sum_{s=1,2} \sum_{\mu \in \mathcal{D}_s} \ell\left(\frac{\boldsymbol{W}_T^{c^\mu} \cdot \boldsymbol{x}^\mu}{\sqrt{d}} + b_T^{c^\mu}, \frac{\boldsymbol{W}_s \cdot \boldsymbol{x}^\mu}{\sqrt{d}} + b_s\right) + \sum_{s=1,2} \frac{\lambda}{2}\left(\sum_{i=1}^d W_{s,i}^2\right) - \frac{\gamma}{2}\|\boldsymbol{W}_1 - \boldsymbol{W}_2\|^2 \quad \text{(B.1)}$$

and we will focus in the following on the cross-entropy loss:

$$\ell(y, q) = -\text{Heav}(y)\log\sigma(q) - (1 - \text{Heav}(y))\log(1 - \sigma(q)) \quad \text{(B.2)}$$

where $\text{Heav}(\cdot)$ is the Heaviside step function, which outputs 1 for positive arguments and 0 for negative ones, and $\sigma(x) = (1 + \exp(-x))^{-1}$ is the sigmoid activation function. The calculation also holds for alternative convex losses, e.g. the Hinge loss or the MSE loss, since the only affected part is the numerical optimisation of the proximal operator, as shown below.

**Teacher partition function**

In the T-M model, the label distribution is non-trivially dependent on the mutual alignment of the shift vector $\boldsymbol{v}$, determining the means of the two Gaussians in the input mixture, and the two teacher vectors $\boldsymbol{W}_T^\pm$. Since we are allowed to fix a Gauge for one of these vectors (compatible with its distribution), we choose for simplicity $\boldsymbol{v} = \boldsymbol{1}$ to be a vector with all entries equal to 1 (still normalized on the sphere of radius $d$). We define the teacher partition function:

$$Z_T = \int d\mu(\boldsymbol{W}_T^+, \boldsymbol{W}_T^-) = \int \prod_{c=\pm} \left[ d\mu\left(\boldsymbol{W}_T^c\right) \delta\left(|\boldsymbol{W}_T^c|^2 - d\right) \delta\left(\boldsymbol{W}_T^c \cdot \boldsymbol{1} - dm_T^c\right) \right] \delta\left(\boldsymbol{W}_T^+ \cdot \boldsymbol{T}_- - d\, q_T\right),$$

where the measures $\mu(\boldsymbol{T}_\pm)$ are in this case assumed to be factorised normal distributions. The Dirac's $\delta$-functions ensure that the geometrical disposition of the model vectors is the one defined by the chosen magnetisations $m_T^\pm$ and the overlap $q_T$, and that the vectors are normalised to the $d$-sphere.

At this point, and throughout this section, we use the integral representation of the $\delta$-function:

$$\delta(x - ad) = \int \frac{d\hat{a}}{2\pi/d} e^{-i\hat{a}\left(\frac{x}{d} - a\right)}, \quad \text{(B.3)}$$

where $\hat{a}$ is a so-called conjugate field that plays a role similar to a Lagrange multiplier, enforcing the constraint contained in the $\delta$-function. We can rewrite:

$$Z_T = \int \prod_{c=\pm} \frac{d\hat{Q}_T^c}{2\pi/d} \int \prod_{c=\pm} \frac{d\hat{m}_T^c}{2\pi/d} \int \frac{d\hat{q}_T}{2\pi/d} e^{d\,\Phi_T\left(\{m_T^\pm, q_T\}, \{\hat{Q}_T^\pm, \hat{m}_T^\pm, \hat{q}_T\}\right)}, \quad \text{(B.4)}$$

where the action $\Phi_T$ represents the entropy of configurations for the teacher that satisfy the chosen geometrical constraints. Given that the components of the teacher vectors are i.i.d., the entropy can be factorised over them. In high dimensions, i.e. when $d \to \infty$, the integral will be dominated by "typical" configurations for the vectors, and the integral $Z_T$ can be computed through a saddle-point approximation. We Wick rotate the fields in order to avoid dealing explicitly with imaginary quantities, and decompose $\Phi_T = g_{Ti} + g_{Ts}$:

$$g_{Ti} = -\left(\sum_c \hat{m}_T^c m_T^c + \sum_c \hat{Q}_T^c + \hat{q}_T q_T\right), \quad \text{(B.5)}$$

$$g_{Ts} = \log \int \mathcal{D}T_+ \int \mathcal{D}T_- \exp\left(\sum_c \hat{Q}_T^c T_c^2 + \sum_c \hat{m}_T^c T_c + \hat{q}_T T_+ T_-\right).$$

After a few Gaussian integrations the computation of the second term yields:

$$g_{Ts} = \frac{\left(1 - 2\hat{Q}_T^-\right)(\hat{m}_T^+)^2 + \left(1 - 2\hat{Q}_T^+\right)(\hat{m}_T^-)^2 + 2\hat{q}_T \hat{m}_T^+ \hat{m}_T^-}{2\left(\left(1 - 2\hat{Q}_T^-\right)\left(1 - 2\hat{Q}_T^+\right) - \hat{q}_T^2\right)} - \frac{1}{2}\log\left(\left(1 - 2\hat{Q}_T^+\right)\left(1 - 2\hat{Q}_T^-\right) - \hat{q}_T^2\right).$$

Now, in order to complete the computation of the partition function $Z_T$, we have impose the saddle point condition for $\Phi_T$, which is realised when the entropy is extremised with respect to the fields we introduced. From the associated saddle point equations one can find two useful identities:

$$1 - m_T^{c\,2} = \frac{\left(1 - 2\hat{Q}_T^{-c}\right)}{\left(\left(1 - 2\hat{Q}_T^-\right)\left(1 - 2\hat{Q}_T^+\right) - \hat{q}_T^2\right)} \tag{B.6}$$

$$q_T - m_T^+ m_T^- = \frac{\hat{q}_T}{\left(\left(1 - 2\hat{Q}_T^-\right)\left(1 - 2\hat{Q}_T^+\right) - \hat{q}_T^2\right)} \tag{B.7}$$

**Free entropy of the learning model**

In this subsection we aim to achieve analytical characterisation of typical learning performance in the T-M, i.e. to describe the solutions of the following optimisation problem:

$$\boldsymbol{W}_1^\star,\ \boldsymbol{W}_2^\star = \underset{\boldsymbol{W}_1,\boldsymbol{W}_2}{\operatorname{argmin}}\ \mathcal{L}(\boldsymbol{W}_1,\boldsymbol{W}_2;\mathcal{D}), \tag{B.8}$$

where $\mathcal{D}$ represents a realisation of the data and $\mathcal{L}(\cdot)$ was defined in Eq. B.1. In typical statistical physics fashion, we can associate this problem with a Boltzmann-Gibbs probability measure, over the possible configurations of the student model parameters:

$$P(\boldsymbol{W}_1,\boldsymbol{W}_2;\mathcal{D}) = \frac{e^{-\beta\mathcal{L}(\boldsymbol{W}_1,\boldsymbol{W}_2;\mathcal{D})}}{Z_W}, \tag{B.9}$$

where the loss $\mathcal{L}$ plays the role of an the energy function, $\beta$ is an inverse temperature and $Z_W$ is the partition function (normalisation of the Boltzmann-Gibbs measure).

Since the loss is convex in the student parameters, when the inverse temperature is sent to infinity, $\beta \to \infty$, the probability measure focuses on the unique minimiser of the loss, representing the solution of the learning problem. In the asymptotic limit $d \to \infty$, the behaviour of this model becomes predictable since the overwhelming majority of the possible dataset realisations (with the same configuration of the generative parameters) will produce solutions with the same macroscopic properties (norm, test performance, etc). We therefore need to consider a self-averaging quantity, which is independent of the specific realisation of the dataset so that the typical learning scenario can be captured.

Thus, we aim to compute the average free-energy:

$$\Phi_W = \lim_{d\to\infty}\lim_{\beta}\frac{1}{\beta d}\left\langle \log Z_W(\boldsymbol{W}_1,\boldsymbol{W}_2;\mathcal{D}_1,\mathcal{D}_2)\right\rangle_{\mathcal{D}_1,\mathcal{D}_2}. \tag{B.10}$$

This type of quenched average is not easily computed because of the log function in the definition. The replica trick, based on the simple identity $\lim_{r\to 0}(x^r - 1)/r = log(x)$, provides a method to tackle this computation. One can replicate the partition function, introducing $r$ independent copies of the original system. Each of them, however, sees the same realisation of the data $\mathcal{D}$ (the "disorder" of the system, in the statistical physics terminology). When one takes the average over $\mathcal{D}$, the $r$ replicas become effectively coupled, and can be intuitively interpreted as i.i.d. samples from the Boltzmann-Gibbs measure of the original problem. At the end of the computation, one takes the analytic continuation of the integer $r$ to the real axis and computes the limit $\lim_{r\to 0}$, re-establishing the logarithm and the initial expression.

We start by computing the replicated volume (product over the $r$ partition functions) $\Omega^r(\mathcal{D})$, which is still explicitly dependent on the sampled dataset:

$$\Omega^r(\mathcal{D}) = \int \frac{d\mu(\boldsymbol{W}_T^+,\boldsymbol{W}_T^-)}{Z_T}\int\prod_{s,a}\left[db_s^a d\boldsymbol{W}_s^a e^{-\frac{\beta\gamma}{2}\|\boldsymbol{W}_1^a - \boldsymbol{W}_2^a\|^2}\prod_{\mu\in\mathcal{D}_s}e^{-\beta\,\ell\left(\frac{\boldsymbol{W}_T^{c\mu}\cdot\boldsymbol{x}^\mu}{\sqrt{d}}+b_T^{c\mu},\frac{\boldsymbol{W}_s^a\cdot\boldsymbol{x}^\mu}{\sqrt{d}}+b_s^a\right)}\right], \tag{B.11}$$

where $s = 1,2$ indexes the two coupled student models and $a = 1,...,r$ is the replica index.

To make progress we have to take the disorder average, i.e. the expectation over the distribution of $\boldsymbol{x}^\mu$ as defined in the T-M model. We can exploit $\delta$-functions in order to replace with dummy variables, $u_\mu$ and $\lambda_\mu^a$, the dot products in the loss and isolate the input dependence in simpler exponential terms:

$$1 = \int \prod_\mu du_\mu\, \delta\left(u_\mu - \frac{\boldsymbol{W}_T^{c^\mu} \cdot \boldsymbol{x}^\mu}{\sqrt{d}}\right) \int \prod_{a,s,\mu\in\mathcal{D}_s} d\lambda_\mu^a \delta\left(\lambda_\mu^a - \frac{\boldsymbol{W}_s^a \cdot \boldsymbol{x}^\mu}{\sqrt{d}}\right) \tag{B.12}$$

$$= \int \prod_\mu \frac{du_\mu d\hat{u}_\mu}{2\pi} e^{i\hat{u}_\mu\left(u_\mu - \sum_{i=1}^d \frac{W_{T,i}^{c^\mu} x_i^\mu}{\sqrt{d}}\right)} \int \prod_{a,s,\mu\in\mathcal{D}_s} \frac{d\lambda_\mu^a d\hat{\lambda}_\mu^a}{2\pi} e^{i\hat{\lambda}_\mu^a\left(\lambda_\mu^a - \sum_{i=1}^d \frac{W_{s,i}^a x_i^\mu}{\sqrt{d}}\right)} \tag{B.13}$$

We can now evaluate the expectation over the input distribution, collecting all the terms where each given input appears. By neglecting terms that vanish in the $N \to \infty$ limit, for each pattern $\mu$ we get:

$$\mathbb{E}_{\boldsymbol{x}^\mu} e^{-i\sum_a \hat{\lambda}_a^\mu \sum_{i=1}^N \frac{W_{s\mu,i}^a x_i^\mu}{\sqrt{d}} - i\hat{u}_\mu \sum_{i=1}^d \frac{W_{T,i}^{c^\mu} x_i^\mu}{\sqrt{d}}} = \tag{B.14}$$

$$= \prod_{i=1}^N e^{-ic^\mu\left(\sum_a \hat{\lambda}_a^\mu \frac{W_{s\mu,i}^a v_i}{d} + \hat{u}^\mu \frac{W_{T,i}^{c^\mu} v_i^\mu}{d}\right)} \mathbb{E}_{z_i^\mu} e^{-i\left(\sum_a \hat{\lambda}_a^\mu \frac{W_{s\mu,i}^a}{\sqrt{d}} + \hat{u}_\mu \frac{W_{T,i}^{c^\mu}}{\sqrt{d}}\right) z_i^\mu}$$

$$= e^{-ic^\mu\left(\sum_a \hat{\lambda}_a^\mu \frac{\sum_i W_{s\mu,i}^a}{d} + \hat{u}^\mu \frac{\sum_i W_{T,i}^{c^\mu}}{d}\right) - \frac{\Delta_{c^\mu}}{2}\left(\sum_{ab} \hat{\lambda}_a^\mu \hat{\lambda}_b^\mu \frac{\sum_i W_{s\mu,i}^a W_{s\mu,i}^b}{d} + 2\hat{u}^\mu \sum_a \hat{\lambda}_a^\mu \frac{\sum_i W_{s\mu,i}^a W_{T,i}^{c^\mu}}{d} + (\hat{u}^\mu)^2 \frac{\sum_i (W_{T,i}^{c^\mu})^2}{d}\right)}. \tag{B.15}$$

To get Eq. B.15, we used the fact that the noise $\boldsymbol{z}^\mu$ is i.i.d. sampled from centred Gaussians of variance determined by the group, and explicitly used our Gauge choice $\boldsymbol{v} = \mathbf{1}$. In this expression, the relevant order parameters of the model appear, describing the overlaps between the student vectors, the shift vector and the teacher vectors. We are thus going to introduce via $\delta$-functions the following parameters:

- $m_s^a = \frac{\boldsymbol{W}_s^a \cdot \mathbf{1}}{d}$, $m_T^c = \frac{\boldsymbol{W}_T^c \cdot \mathbf{1}}{d}$: magentisations in the direction of the $+$ group centre of the students and the teachers.

- $q_s^{ab} = \frac{\sum_i w_{si}^a w_{si}^b}{d}$: self-overlap between different replicas of each student.

- $R_{sc}^a = \frac{\sum_i W_{s,i}^a W_{T,i}^c}{d}$: overlap between student and teacher vectors.

- $q_T^c = \frac{\sum_i T_{ci}^2}{d}$: norm of the teacher vectors (equal to 1 by assumption).

After the introduction of these order parameters (via the integral representation of the $\delta$-function) the replicated volume can be expressed as:

$$\Omega^d = \int \prod_{s,a} \frac{dm_s^a d\hat{m}_s^a}{2\pi/d} \int \prod_{sc,a} \frac{dR_{sc}^a d\hat{R}_{sc}^a}{2\pi/d} \int \prod_{s,ab} \frac{dq_s^{ab} d\hat{q}_s^{ab}}{2\pi/d} \int \prod_c db_c^a G_I^d G_S^d \prod_{sc} G_E(s,c)^{\alpha_{c,s}d} \tag{B.16}$$

where $\alpha_{c,s}N$ indicates the number of patterns from group $c$ contained in the data slice $\mathcal{D}_s$ given to student $s$. We also introduced the interaction, the entropic and the energetic terms:

$$G_I = \exp\left(-\sum_{s,a} \hat{m}_s^a m_s^a - \sum_{s,ab} \hat{q}_s^{ab} q_s^{ab} - \sum_{sc,a} \hat{R}_{sc}^a R_{sc}^a\right) \tag{B.17}$$

$$G_S = \int \prod_c \mathcal{D}T_c \exp\left(\sum_c \hat{Q}_T^c T_c^2 + \sum_c \hat{m}_T^c T_c + \hat{q}_T T_+ T_-\right)$$

$$\times \int \prod_{s,a} d\mu\left(w_s^a\right) e^{-\beta\gamma(w_1^a - w_2^a)^2} \exp\left(\sum_{s,a} \hat{m}_s^a w_s^a + \sum_{s,ab} \hat{q}_s^{ab} w_s^a w_s^b + \sum_{sc,a} \hat{R}_{sc}^a w_s^a T_c\right) \tag{B.18}$$

$$G_E(s,c) = \int \frac{du d\hat{u}}{2\pi} e^{iu\hat{u}} \int \prod_a \left(\frac{d\lambda^a d\hat{\lambda}^a}{2\pi} e^{i\lambda^a \hat{\lambda}^a}\right) e^{-\frac{\Delta^c}{2}\sum_{ab} \hat{\lambda}_a \hat{\lambda}_b q_s^{ab} - \Delta^c \hat{u} \sum_a \hat{\lambda}_a R_{sc}^a - \frac{\Delta^c}{2}(\hat{u})^2}$$

$$\times \prod_a e^{-\beta\,\ell(u + cm_T^c + b_T^c,\, \lambda^a + cm_s^a + b_s^a)} \tag{B.19}$$

The shorthand notation $\mathcal{D}x = \frac{e^{-\frac{x^2}{2}}}{\sqrt{2\pi}}$ is used to indicate a normal Gaussian measure. Note that, after the factorization in the $G_S$, the variables $T_c$ and $w_s^a$ denote a component of the vectors $\boldsymbol{W}_T^c$ and $\boldsymbol{W}_s^a$ respectively.

**Replica symmetric ansatz.** To make further progress, we have to make an assumption for the structure of the introduced order parameters. Given the convex nature of the optimisation objective B.1, the simplest possible ansatz, the so-called replica symmetric (RS) ansatz, is fortunately exact. Replica symmetry introduces a strong constraint for the overlap parameters, requiring the $r$ replicas of the students to be indistinguishable and the free entropy to be invariant under their permutation. Mathematically, the RS ansatz implies that:

- $m_s^a = m_s$ for all $a = 1, ..., r$ (same for the conjugate)

- $R_{sc}^a = R_{sc}$ for all $a = 1, ..., r$ (same for the conjugate)

- $q_s^{ab} = q_s$ for all $a > b$, $q_s^{ab} = Q_s$ for all $a = b$ (same for the conjugate)

- $b_s^a = b_s$ for all $a = 1, ..., r$

Moreover, since we want to describe the minimisers of the loss, we are going to take the $\beta \to \infty$ limit in the Gibbs-Boltzmann measure. The replicas, which represent independent samples from it, will collapse on the unique minimum. This is represented by the following scaling law with $\beta$ for the order parameters, which will be used below:

$$Q - q = \delta q/\beta; \quad \hat{Q} - \hat{q} = -\beta\delta\hat{q}; \quad \hat{q} \sim \beta^2 \hat{q}; \quad \hat{m} \sim \beta\hat{m}; \quad \hat{R} \sim \beta\hat{R} \tag{B.20}$$

**Interaction term.** We now proceed with the calculation of the different terms in B.16, where we can substitute the RS ansatz. In the interaction term, neglecting terms of $\mathcal{O}(n^2)$, we get:

$$G_i = \exp\left(-n\left(\sum_s \left(\hat{m}_s m_s + \sum_c \hat{R}_{sc} R_{sc} + \frac{\hat{Q}_s Q_s}{2} - \frac{\hat{q}_s q_s}{2}\right)\right)\right) \tag{B.21}$$

In the $\beta \to \infty$ limit the expression becomes:

$$\log(G_i)/d = g_i = -\beta\left(\sum_s \left(\hat{m}_s m_s + \sum_c \hat{R}_{sc} R_{sc} + \frac{1}{2}\left(\hat{q}_s \delta q_s - \delta\hat{q}_s q_s\right)\right)\right) \tag{B.22}$$

**Entropic term**

In the entropic term the computation is more involved, due to the couplings between the Gaussian measures for the teachers and for those of the students. We substitute the RS ansatz in expression B.18 to get:

$$G_S = \int \mathcal{D}T_+ \int \mathcal{D}T_- \exp\left(\sum_c \hat{Q}_T^c T_c^2 + \sum_c \hat{m}_T^c T_c + \hat{q}_T T_+ T_-\right) \int \prod_{s,a} d\mu\left(w_s^a\right) e^{-\frac{\gamma}{2}(w_1^a - w_2^a)^2}$$

$$\times \prod_s \exp\left(\hat{m}_s \sum_a w_s^a + \frac{1}{2}\left(\hat{Q}_s - \hat{q}_s\right)\sum_a (w_s^a)^2 + \frac{1}{2}\hat{q}_s \left(\sum_a w_s^a\right)^2 + \sum_c \hat{R}_{sc} \sum_a w_s^a T_c\right) \tag{B.23}$$

We perform a Hubbard-Stratonovich transformation to remove the squared sum in the previous equation, introducing the Gaussian fields $z_s$. Then, we rewrite coupling term between the teachers as $\hat{q}_T T_+ T_- = \frac{\hat{q}_T}{2}(T_+ + T_-)^2 - \frac{\hat{q}_T}{2}(T_+^2 + T_-^2)$, and perform a second Hubbard-Stratonovich transformation, with field $\tilde{z}$, to remove the explicit coupling between $T_+$ and $T_-$. Similarly, the elastic coupling between the students can be turned into a linear term with fields $z_{12}^a$:

$$
= \int \mathcal{D}\tilde{z} \int \prod_s \mathcal{D}z_s \int \frac{dT_c}{\sqrt{2\pi}} \exp\left( -\frac{1}{2}\sum_c \left(1 - 2\hat{Q}_T^c + \hat{q}_T\right) T_c^2 + \sum_c \left(\hat{m}_T^c + \sqrt{\hat{q}_T}\tilde{z}\right) T_c \right) \int \prod_a \mathcal{D}z_{12}^a
$$

$$
\times \int \prod_{s,a} d\mu\left(w_s^a\right) \prod_s \exp\left( \frac{1}{2}\left(\hat{Q}_s - \hat{q}_s\right)\sum_a (w_s^a)^2 + \left(\hat{m}_s + \sum_c \hat{R}_{sc}T_c + \sqrt{\hat{q}_s}z_s + is\sqrt{\gamma}z_{12}^a\right)\sum_a w_s^a \right) \qquad \text{(B.24)}
$$

After rescaling the variances of the teacher measures and centring them, one can factorise over the replica index and take the $r \to 0$ limit, obtaining the following expression for $g_S = \log G_S/d$:

$$
g_S = A + \int \prod_s \mathcal{D}z_s \int \prod_c \mathcal{D}T_c \int \mathcal{D}\tilde{z} \log \int \mathcal{D}z_{12} \int \prod_s d\mu\left(w_s\right) \exp\left( \frac{1}{2}\left(\hat{Q}_s - \hat{q}_s\right) w_s^2 + B_s w_s \right) \qquad \text{(B.25)}
$$

where:

$$
A = \frac{\sum_c \hat{m}_T^{c\,2}\left(1 - 2\hat{Q}_T^{-c}\right) + 2\hat{q}_T\left(\sum_c \hat{m}_T^c\right)^2}{2\left(\left(1 - 2\hat{Q}_T^+\right)\left(1 - 2\hat{Q}_T^-\right) - \hat{q}_T^2\right)} \qquad \text{(B.26)}
$$

$$
B_s = b_s\left(T_\pm, z_\pm, \tilde{z}, z_s\right) + is\sqrt{\gamma}z_{12} \qquad \text{(B.27)}
$$

$$
b_s = \hat{m}_s + \sqrt{\hat{q}_s}z_s + \sum_c \left[ m_T^c \hat{R}_{sc} + \frac{\hat{R}_{sc}}{\sqrt{\left(1 - 2\hat{Q}_T^c + \hat{q}_T\right)}}T_c + \frac{\sqrt{\hat{q}_T}}{\sqrt{1 - \sum_{c'} \frac{\hat{q}_T}{\left(1 - 2\hat{Q}_T^{c'} + \hat{q}_T\right)}}} \frac{\hat{R}_{sc}}{\left(1 - 2\hat{Q}_T^c + \hat{q}_T\right)}\tilde{z} \right]
$$
$$
\text{(B.28)}
$$

In the $\beta \to \infty$ limit, and considering the $L_2$-regularisation on the student weights $d\mu\left(w\right) = \frac{dw}{\sqrt{2\pi}}e^{-\frac{\beta\lambda}{2}w^2}$ we get:

$$
g_S = A + \int \prod_s \mathcal{D}z_c \int \prod_c \mathcal{D}T_c \int \mathcal{D}\tilde{z} \log \int \mathcal{D}z_{12} \exp\left( \sum_s \max_{w_s}\left( -\frac{\lambda + \delta\hat{q}_s}{2}w_s^2 + B_s w_s \right) \right) \qquad \text{(B.29)}
$$

and the maximisation gives:

$$
w_s^\star = \frac{B_s}{\left(\lambda + \delta\hat{q}_s\right)}; \quad \max_{w_s}\left( -\frac{\lambda + \delta\hat{q}_s}{2}w_s^2 + B_s w_s \right) = \frac{B_s^2}{2\left(\lambda + \delta\hat{q}_s\right)} \qquad \text{(B.30)}
$$

Substituting the above described scaling laws for the order parameters in the $\beta \to \infty$ limit one finds that the $A$ term becomes sub-dominant and can be ignored. The remaining steps are quite tedious, but the procedure to obtain the final result for the entropic channel is straightforward:

- Expand the sums in Eq.B.29.

- Perform the $z_{12}$ Gaussian integration and take the log of the result.

- Identify the terms that have even powers in the Hubbard-Stratonovich Gaussian fields and in the teacher variables. The Gaussian integrations will kill all the remaining cross terms, so they can be ignored.

- Perform the remaining Gaussian integrations.

- Use identities B.6 and B.7 to remove the dependence on the conjugate fields appearing in the Teacher measure and only retain a dependence on $m_T^c$, $Q_T^c$, and $q_T$.

The final expression reads:

$$g_S = \frac{\beta}{2\left(\prod_s(\lambda+\gamma+\delta\hat{q}_s)-\gamma^2\right)}\left[\left(\sum_s\left(\hat{m}_s+\sum_s m_T^c\hat{R}_{sc}\right)^2(\lambda+\gamma+\delta\hat{q}_{\neg s})+2\gamma\prod_s\left(\hat{m}_s+\sum_c m_T^c\hat{R}_{sc}\right)\right)\right.$$
(B.31)

$$+\left(\sum_s\hat{q}_s\left(\lambda+\gamma+\delta\hat{q}_{\neg s}\right)\right)+\left(\sum_c\left(1-m_T^{c\;2}\right)\left(\sum_s\hat{R}_{sc}^2\left(\lambda+\gamma+\delta\hat{q}_{\neg s}\right)+2\gamma\prod_s\hat{R}_{sc}\right)\right)$$
(B.32)

$$\left.+\left(2\left(q_T-m_T^+m_T^-\right)\left(\sum_s\left(\prod_c\hat{R}_{sc}\left(\lambda+\delta\hat{q}_{\neg s}\right)\right)+\gamma\left(\prod_c\left(\sum_s\hat{R}_{sc}\right)\right)\right)\right)\right]$$
(B.33)

where the notation $\neg s$ denotes the other student index with respect to the one used in the corresponding sum or product.

**Energetic term.** We can compute the energetic channel for a generic student $s$ and a generic data group $c$. Each term will be multiplied by $\alpha_{c,s}$, determining the fraction of inputs from group $c$ in the dataset $\mathcal{D}_s$ of student $s$. For simplifying the notation in this subsection we drop the indices $s, c$, with the understanding that the all the order parameters, and model parameters, appearing in the following expressions are those corresponding to a specific pair of these indices.

Substituting the RS ansatz in Eq. B.19 we get:

$$G_E = \int\frac{du d\hat{u}}{2\pi}e^{iu\hat{u}}\int\prod_a\left(\frac{d\lambda^a d\hat{\lambda}^a}{2\pi}e^{i\lambda^a\hat{\lambda}^a}\right)e^{-\frac{\Delta}{2}\sum_{ab}\hat{\lambda}_a\hat{\lambda}_b q-\Delta\hat{u}R\sum_a\hat{\lambda}_a-\frac{\Delta}{2}(\hat{u})^2 q_T}$$
(B.34)

$$\times\prod_a e^{-\beta\,\ell\left(u+c\tilde{m}+\tilde{b},\lambda^a+cm+b\right)}$$
(B.35)

We can start by evaluating the Gaussian in $\hat{u}$, then performing a Hubbard-Stratonovich transformation, with field $z$, to remove the squared sums on the replica index. Following up with the Gaussian integration in $\hat{\lambda}$ we find that the argument of the integrations factorises over the replica index. Up to first order in $r$ when $r\to 0$, we find for $g_E = \log G_E/d$:

$$g_E = \int\mathcal{D}z\int\mathcal{D}u\log\int\mathcal{D}\lambda e^{-\beta\,\ell\left(\sqrt{\Delta q_T}u+c\tilde{m}+\tilde{b},\sqrt{\Delta(Q-q)}\lambda+\frac{\sqrt{\Delta}R}{\sqrt{q_T}}u+\sqrt{\Delta\frac{(q-R^2)}{q_T}}z+cm+b\right)}$$
(B.36)

and in the the $\beta\to\infty$ limit we can solve the integral by saddle-point:

$$\log\int\mathcal{D}\lambda e^{-\beta\,\ell\left(\sqrt{\Delta q_T}u+c\tilde{m}+\tilde{b},\sqrt{\Delta(Q-q)}\lambda+\frac{\sqrt{\Delta}R}{\sqrt{q_T}}u+\sqrt{\Delta\frac{(q-R^2)}{q_T}}z+cm+b\right)}=-\beta M$$
(B.37)

with:

$$M = \min_\lambda\frac{\lambda^2}{2}+\ell\left(\sqrt{\Delta q_T}u+c\tilde{m}+\tilde{b},\sqrt{\Delta\delta q}\lambda+\frac{\sqrt{\Delta}R}{\sqrt{q_T}}u+\sqrt{\Delta\frac{(q-R^2)}{q_T}}z+cm+b\right)$$
(B.38)

To simplify further, we can shift $\frac{\sqrt{\Delta}R}{\sqrt{q_T}}u+\sqrt{\Delta\frac{(q-R^2)}{q_T}}z\to\sqrt{\Delta q}z'$. Then, given the definition of the logistic loss B.2, we can split the $u$ integration over the intervals $\sqrt{\Delta q_T}u+c\tilde{m}_c > 0$ and $\sqrt{\Delta q_T}u+c\tilde{m}_c < 0$ and eventually get (re-establishing the $s, c$ indices):

$$g_E(s,c) = \sum_y\int\mathcal{D}z H\left(-y\frac{q_s\frac{cm_T^c+b_T^c}{\sqrt{1}}+\sqrt{\Delta_c}R_{sc}z}{\sqrt{\Delta_c\left(q_s-R_{sc}^2\right)}}\right)M_E(y,s,c)$$
(B.39)

Where $H(x) = \frac{1}{2}\operatorname{erfc}(x/\sqrt{2})$ is the Gaussian tail function and we defined the proximal:

$$M_E(y,s,c) = \max_\lambda -\frac{\lambda^2}{2}-\ell\left(y,\sqrt{\Delta_c\delta q_s}\lambda+\sqrt{\Delta_c q_s}z+cm_s+b_s\right)$$
(B.40)

Note that this simple 1D optimisation problem has to be solved numerically in correspondence of each point evaluated in the integral.

The reweighing strategy is easily embedded in this calculation by explicitly changing the definition of $\ell$, adding a different weight $\mathcal{W}_{c,y}$ for each combination of label and group membership. Defining a one-hot encoding vector for the teacher-produced label, $Y \in \mathbb{R}^2$, and a output probability (constructed from the sigmoid function) for the student, $P(\hat{Y})$, the reweighed cross-entropy loss can be written as:

$$\mathcal{L}(\mathcal{D}) = \sum_{c=\pm} \sum_{y=0,1} (\mathcal{W})_{(c,y)} Y_y \log P(\hat{Y}_y). \tag{B.41}$$

For the sake of simplicity we reduced the degrees of freedom to two, parameterising these weights as:

$$\mathcal{W} = 2 \begin{pmatrix} w_+ w_1 & w_+(1-w_1) \\ (1-w_+)w_1 & (1-w_+)(1-w_1) \end{pmatrix} \tag{B.42}$$

where $w_+, w_1 \in [0,1]$ can be used to increase the relative weight of a misclassification errors in the group $+$ and label 1 respectively.

Different losses could be chosen instead of the cross-entropy and, again, only the numerical optimisation of the proximal would be affected.

**Saddle-point of the free-entropy**

We thus have found that the free-entropy $\Phi_W$ can be written as a simple function of few scalar order parameters. In the high-dimensional limit, the integral in B.16 is dominated by the typical configuration of the order parameters, which is found by extremising the free-entropy with respect to all the overlap parameters:

$$\Phi_W = \underset{o.p.}{\mathrm{extr}} \left\{ g_I + g_S + \sum_{s,c} \alpha_{s,c}\, g_E(s,c) \right\} \tag{B.43}$$

The saddle-point is typically found by fixed-poimnt iteration: setting each derivative, with respect to the order parameters, to zero returns a saddle-point condition for the conjugate parameters, and vice-versa.

The fixed-point is uniquely determined by the value of the generative parameters, $m_T^{\pm}$ and $q_T$, and the pattern densities $\alpha_{s,c}$. In the main text, for simplicity, we parameterise $\alpha_{s,c}$ through the fraction $\eta$, which represents the percentage of patterns from group $+$ assigned to the first student model.

The special case of a single student model is obtained from this calculation by setting $\gamma = 0$ and assigning all the inputs in the first dataset $\mathcal{D}_1$.

**Test accuracy.** All the performance assessment metrics employed in this paper can be derived from the confusion matrix, which measures the TP, FP, TN, FN rates on new samples from the T-M. These quantities can be evaluated analytically and are easily expressed as a function of the saddle-point order parameters obtained in the previous paragraphs.

Suppose we obtain a new data point with label $y$ from group $c$, then probability of obtaining an output $\hat{y}$ from the trained model $s$ is given by:

$$P\left(Y=y,\hat{Y}=\hat{y}\right) = \mathbb{E}_{\mathbf{x}(c)} \left\langle \Theta\left(y\left(\frac{\mathbf{W}_T^c \cdot \mathbf{x}(c)}{\sqrt{d}} + \tilde{b}\right)\right) \Theta\left(\hat{y}\left(\frac{\mathbf{W}_s \cdot \mathbf{x}(c)}{\sqrt{d}} + b\right)\right) \right\rangle_{\mu(\boldsymbol{W}_T, \boldsymbol{W})} \tag{B.44}$$

$$= \mathbb{E}_{\mathbf{x}(c)} \left\langle \int \frac{du d\hat{u}}{2\pi} e^{i\hat{u}\left(u - \sum_{i=1}^d \frac{W_{T,i} x_i}{\sqrt{d}}\right)} \int \frac{d\lambda d\hat{\lambda}}{2\pi} e^{i\hat{\lambda}\left(\lambda - \sum_{i=1}^d \frac{W_i x_i}{\sqrt{d}}\right)} \right\rangle \Theta\left(y\left(u+\tilde{b}\right)\right) \Theta\left(\hat{y}\left(\lambda+b\right)\right) \tag{B.45}$$

where, following the same lines as in the free-entropy computation, we used $\delta$-functions to extract the dependence on the input, to facilitate the expectation:

$$\mathbb{E}_{\boldsymbol{x}(c)} \left\langle e^{-i\hat{\lambda}\frac{\boldsymbol{W}_s \cdot \boldsymbol{x}(c)}{\sqrt{d}} - i\hat{u}\frac{\boldsymbol{W}_T \cdot \mathbf{x}(c)}{\sqrt{d}}} \right\rangle \tag{B.46}$$

$$= e^{-ic(\hat{\lambda}m+\hat{u}\bar{m})}e^{-\frac{\Delta}{2}(\hat{\lambda}^2 Q+2\hat{u}\hat{\lambda}R+\hat{u}^2)}. \tag{B.47}$$

We have substituted the overlaps that come out of the average with their typical values in the Boltzmann-Gibbs measure of the T-M. Note that we can substitute $q = Q$ since in the $\beta \to \infty$ limit they are equal up to the first order.

The Gaussian integrals can be computed and one gets the final expression:

$$P\left(Y = y, \hat{Y} = \hat{y}\right) = \int_{-\infty}^{\infty} \mathcal{D}u\Theta\left(y\left(\sqrt{\Delta_c}u + cm_T^c + b_T^c\right)\right) H\left(-\hat{y}\frac{\sqrt{\Delta_c}R_{sc}u + cm_s + b_s}{\sqrt{\Delta_c\left(q_s - R_{sc}^2\right)}}\right) \tag{B.48}$$

Similarly, one can also obtain e.g. the label 1 frequency:

$$P\left(Y = 1\right) = \rho H\left(-\frac{m_T^+ + b_T^+}{\sqrt{\Delta_+}}\right) + (1 - \rho) H\left(\frac{m_T^- - b_T^-}{\sqrt{\Delta_-}}\right) \tag{B.49}$$

and the generalisation error:

$$\epsilon_g = \int_{-\infty}^{\infty} \mathcal{D}u H\left(\text{sign}\left(\left(\sqrt{\Delta_c}u + cm_T^c + b_T^c\right)\right)\frac{\sqrt{\Delta_c}R_{sc}u + cm_s + b_s}{\sqrt{\Delta_c\left(q_s - R_{sc}^2\right)}}\right). \tag{B.50}$$

### B.1 Parameters used in the figures

The following list contains the parameters of the T-M model used to plot the figures of the paper.

- Fig. 1d: $\Delta_+ = 0.5, \Delta_- = 2_0.5, \alpha = 2.5, q_T - 0.2$.

- Fig. **??**a (upper): $m_\pm = 0.2,\ \alpha = 0.5, \Delta_+ = 0.5, \Delta_- = 0.5, b_+ = 0, b_- = 0$.

- Fig. **??**a (lower): $\alpha = 0.5,\ \Delta_+ = 0.5, \Delta_- = 0.5, b_+ = 0, b_- = 0$.

- Fig. **??**b: $\alpha = 0.5, q_T = 1, m = 0.5, b_+ = 0, b_- = 0$.

- Fig. **??**c: $\rho = 0.1,\ m = 0.2, \Delta_+ = 0.5, \Delta_- = 0.5, b_+ = 0, b_- = 0$.

- Fig. 6a: $\rho = 0.1, q_T = 0.8, \Delta_+ = 2.0, \Delta_- = 0.5, \alpha = 0.5, m_+ = 0.3, m_- = 0.1, b_+ = 0.5, b_- = 0.5$.

## C Real data validation

In the next two sections, we demonstrate the ability of the Teacher-Mixture model to mimic unfairness scenarios in real-world applications. In particular, we perform this validation through a set of numerical experiments on the CelebA dataset Liu et al. (2015). This dataset consists of a collection of face images of celebrities, equipped with metadata indicating the presence of specific attributes in each picture. As can be seen in Fig. C.3, the consistent amount of these attributes allows to explore many possible learning scenarios in unfairness conditions. This feature of CelebA together with its size and the high-dimensional nature of face pictures, makes it a good candidate for validating the Teacher-Mixture model on real datasets. Moreover, as shown in Fig. C.2 through a PCA clustering, the different sub-populations associated to a given CelebA attribute are overlapping and hard to disentangle. This situation precisely corresponds to the high-noise regime the Teacher-Mixture model is meant to describe. Interestingly, the picture emerging from the simulations on CelebA turned out to be quite general and further extendable to lower-dimensional datasets such as the Medical Expenditure Panel Survey (MEPS) dataset Blewett et al. (2021). More details on both datasets are discussed in Sec. C.2 and Sec. C.3. Here we provide a general overview on the experimental framework applied to CelebA.

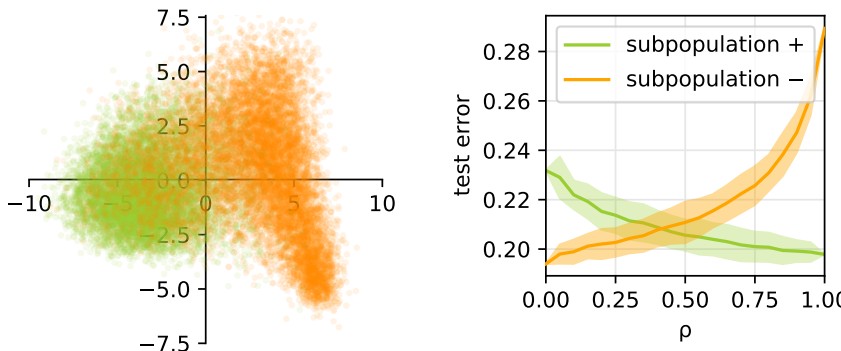

Figure C.1: **Relative representation and bias**. Numerical experiments on a sub-sample of the CelebA dataset. *(Left)* A 2D projection of the pre-processed dataset, obtained from PCA, where the colours represent the two sub-populations. *(Right)* Per community test error, as the fraction of samples from the two subpopulations is varied (dataset dimension is fixed).

### C.1 Model motivation

We construct a dataset by sub-sampling CelebA and by preprocessing the selected images through an Xception network Chollet (2017) trained on ImageNet Deng et al. (2009). As depicted in the scatter plot in Fig. C.1, the first two principal components of the obtained data clearly reveal a clustered structure. Many attributes contained in the metadata are highly correlated with the split into these two sub-populations. For example, in the figure we colour the points according to the attribute "Wearing_Lipstick". Now, suppose we are interested in predicting a different target attribute, which is not as easily determined by just looking at the group membership, e.g. "Wavy_Hair"[2]. What happens to the model accuracy if one alters the *relative representation* of the two groups, e.g. when one varies the fraction of points that belong to the orange group?

The right panel of Fig. C.1 shows the outcome of this experiment. As we can see from the plot, the fact that a group is under-represented induces a gap in the generalisation performance of the model when evaluated on the different sub-populations. The presence of a gap is a clear indicator of unfairness, induced by an implicit bias towards the over-represented group.

Many factors might play a role in determining and exacerbating this phenomenon. This is precisely why designing a general recipe for a fair / unbiased classifier is a very challenging, if solvable, problem. Some bias inducing factors are linked to the sampling quality of the dataset, as in the case of the overall number of datapoints and the balance between the sub-populations frequencies. Other factors are controlled by the different degree of variability in the input distributions of each group. In other cases the imbalance is hidden and can only be recognised by looking at the joint distribution of inputs and labels. For example, the balance between the positive/negative labels might differ among the groups and may be strongly correlated with the group membership. Even similar individuals with different group memberships might be labelled differently. The present work aims at modelling the data structure observed in these types of experiments, to obtain detailed understanding of the various sources of bias in these problems.

### C.2 Additional details on the CelebA experiments

The CelebA dataset is a collection of 202.599 face images of various celebrities, accompanied by 40 binary attributes per image (for instance, whether a celebrity features black hairs or not) Liu et al. (2015). To obtain the results presented in the main text we apply the following pre-processing pipeline: We first downsample CelebA up to 20.000 images. Notice that this is done with the purpose of considering settings with limited amount of available data. Indeed, as we have seen in the main manuscript, data scarcity is one of the main bias-inducing ingredients. We are thus not interested to consider the entire CelebA dataset, especially for simple classification tasks like the one described in the main text. By exploiting the deep learning framework provided by Tensorflow Abadi et al. (2015), we then pre-process the dataset using the features extracted from an Xception convolutional network Chollet (2017) pre-trained on Imagenet Deng et al. (2009). Finally, we collect the extracted features together with the associated binary attributes in a json file.

---

[2]To be mindful on the Ethical Considerations of using the CelebA datast, we don't use protected attributes like binary genders and age Denton et al. (2019)

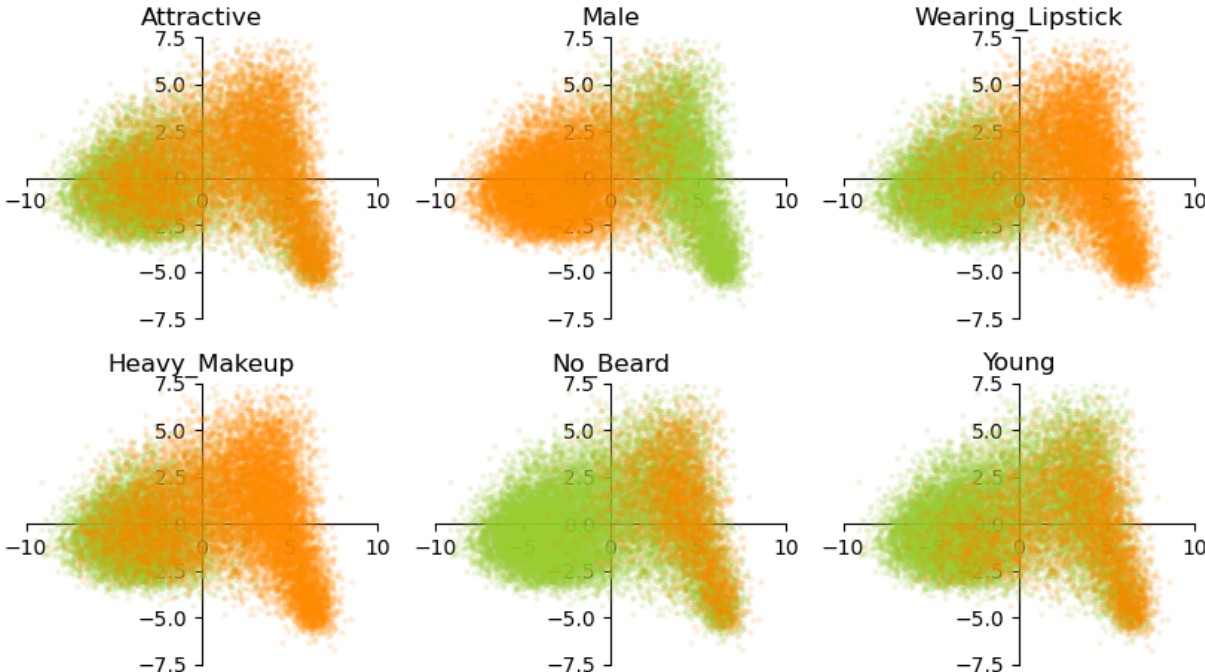

Figure C.2: **Clustering CelebA according to attributes.** We show 6 of the 40 attributes in CelebA demonstrating a neat clustering.

By applying PCA on the pre-processed dataset, we observe a clustering structure in the data when projected to the space of the PCA principal components. The clusters appear to reflect a natural correspondence with the binary attributes associated to each input data point, however this is not a general implication and many datasets show clustering with a non interpretable connection to the attributes. The clusters can be clearly seen in Fig. C.2, where we use colours to show whether a celebrity features a given attribute (green dots) or not (orange dots). In the plot, the axes correspond to the directions traced by the two PCA leading eigenvectors. As we can see from Fig. C.2, the two sub-populations are overlapping and hard to disentangle. This situation precisely corresponds to the high-noise regime the T-M model is meant to describe. Among the various clustering depicted in Fig. C.2, we decided to disregard those corresponding to ethically questionable attributes, such as "Attractive", "Male" or "Young". Finally, we chose as sensitive attribute – determining the membership in the subpopulations – the "Wearing_Lipstick" feature since it gives a more homogeneous distribution of the data points in the two clusters.

Anyone of the other attributes can be considered as a possible target, and thus be used to label the data points. The final pre-processing step consist in downsampling further the data in order to have the same ratio of 0 and 1 labels in the two subpopulations. This step helps mitigating bias induced by the different ratio of label in the two subpopulations and simplifies the identification of the other sources of bias. The general case can be addressed in the T-M model, in Sec. D we comment more on the bias induced by different label ratios.

As Fig. C.3 illustrates, there is a large number of possible outcomes concerning the behaviour of the test error as a function of the relative representation. Indeed, as we have seen in the main text, the presence and the position of the crossing point strictly depends on both the cluster variances and the amount of available data. Despite all these behaviours are fully reproducible in the T-M model by means of its corresponding parameters, we here decided to chose the "Wavy_Hair" as target feature because it shows a nicely symmetric profile of the test error that is more suitable for illustration purposes. To get the learning curves in Fig. C.3, we train a classifier with logistic regression and $L_2$-regularisation. In particular, we use the LogisticRegression class from scikit-learn Pedregosa et al. (2011). This class implements several logistic regression solvers, among which the *lbfgs* optimizer. This solver implements a second order gradient descent optimization which can consistently speed-up the training process. The training algorithm stops either if the maximum component of the gradient goes below a certain threshold, or if a maximum number of iterations is reached. In our case, we set the threshold at $1e - 15$ and the maximum number of iterations to $10^5$. The parameter *penalty* of the

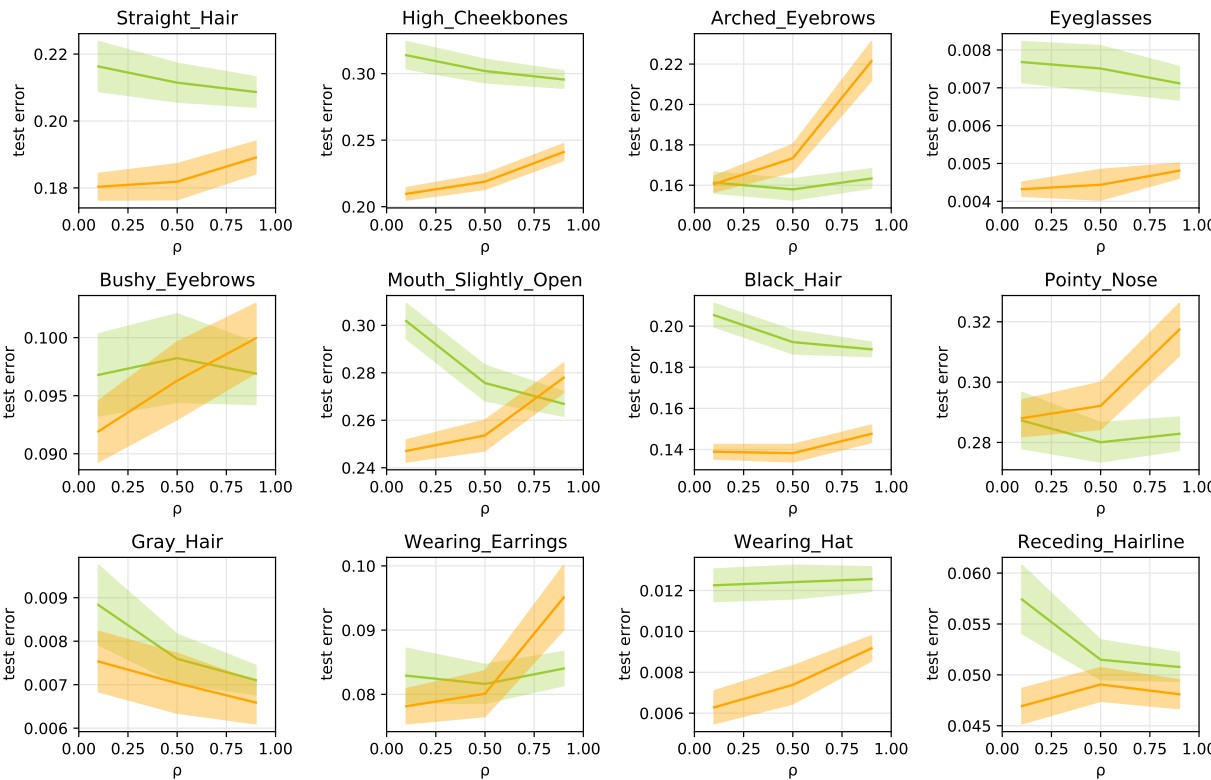

Figure C.3: **Relative representation across attributes.** The panels show the generalisation error depending on the relative representation in different attributes. The sub-populations + (green) − (orange) are obtained splitting according to the attribute "Wearing_Lipstick". The simulations are averaged over 100 samples.

LogisticRegression class is a flag determining whether an $L_2$-regularisation needs to be added to the training or not. The C hyper-parameter corresponds instead to the inverse of the regularisation strength. In our experiments, we chose the value of the regularisation strength by cross-validation in the interval $(10^{-3}, 10^3)$ with 30 points sampled in logarithmic scale.

## C.3   Other datasets

The observations made on the CelebA dataset are quite general and can be further extended to lower-dimensional datasets. As example of this, we considered the Medical Expenditure Panel Survey (MEPS) dataset. This is a dataset containing a large set of surveys which have been conducted across the United States in order to quantify the cost and use of health care and health insurance coverage. The dataset consists of about 150 features, including sensitive attributes, such as age or medical sex, as well as attributes describing the clinical status of each patient. The label is instead binary and measures the expenditure on medical services of each individual, assessing whether the total amount of medical expenses is below or above a certain threshold. As it can be seen in Fig. C.4, the behaviour is qualitatively similar to the one already observed in the CelebA dataset of celebrity face images. Indeed, even in this case, PCA shows the presence of two distinct clusters when considering the age as the sensitive attribute and then splitting the dataset in two sub-populations, according to the middle point of the age distribution. Moreover, the generalisation error per community exhibits a crossing according to the relative representation.

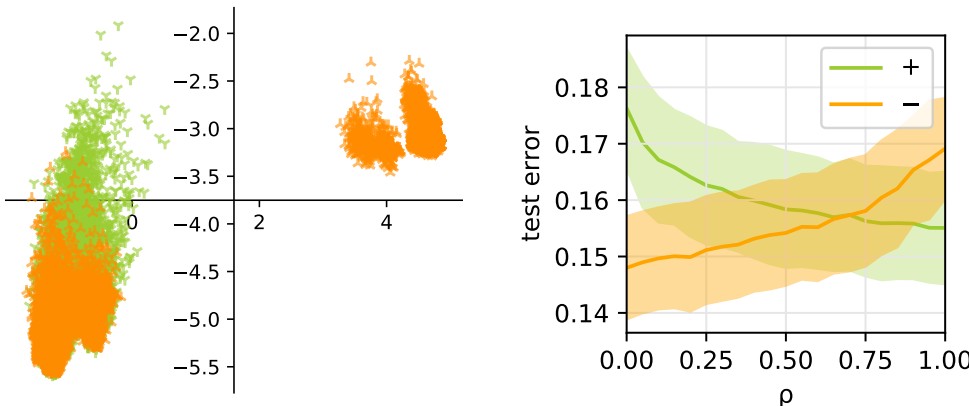

Figure C.4: **MEPS dataset.** (Left) Clustering in the MEPS dataset, according to be above or below the average age. (Right) Crossing of the generalisation error as the relative representation $\rho$ is changed. The simulations are averaged over 100 samples.

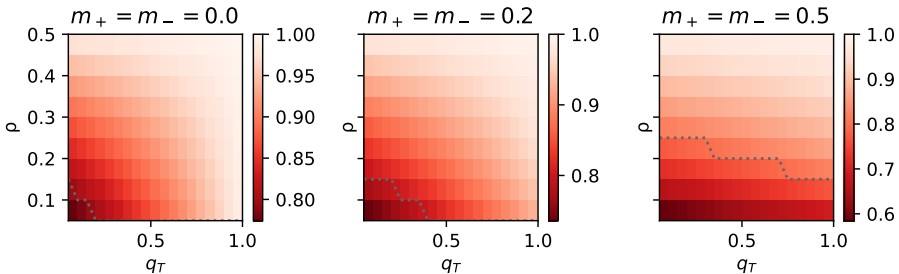

Figure D.1: **Bias with two different rules to be learned.** The three phase diagrams give the DI depending on $\rho$ (y-axis) and $q_T$ (x-axis). Moving from the left panel to the right panel $m_+$ and $m_i$ are increased. The other parameters are: $\alpha = 0.5, \Delta_+ = 0.5, \Delta_- = 0.5, b_+ = 0, b_- = 0$.

# D  Exploration of the parameter space

## D.1  Supporting results

This section presents supporting results on the sources of bias. In Fig. D.1, we re-propose the the study of the disparate impact (DI) depending on the relative representation $\rho$ and the rule similarity $q_T$, paying close attention to the role of the group-label correlation $m_+$, $m_-$. Interestingly, if $m_+ = m_- = 0$, when the rules become identical ($q_T = 1$) the bias is removed. However if $m_+ = m_- \neq 0$ this is no longer true. This shows once again that it is not sufficient for a classifier to be able of reproducing the rule, as bias can appear in reason of other concurring factors.

The main difference with respect to the case with $q_T \neq 1$ is that, if $q_T = 1$, increasing the amount of training data can be a solution. In fact, bias at $q_T = 1$ is due to overfitting with respect ot the largest sub-population, and this effect can be cured by increasing in $\alpha$. This is illustrated in Fig. D.2, that extends the figure of the main text showing the effect of $\alpha$. Moving from left to right, $\alpha$ increases and the area where the 80% rule is violated shrinks down.

The results shown until this point are agnostic with respect to the relative fraction of labels inside the sub-populations. When this quantity is strongly varied across the groups, it can contribute to an additional source of bias, especially if combined with a small relative representation. Indeed, the classifier can simply bias its prediction towards the most likely outcome reaching an accuracy that apparently exceeds random guessing, without effectively doing any informed prediction. Many factors play a role in deciding the relative

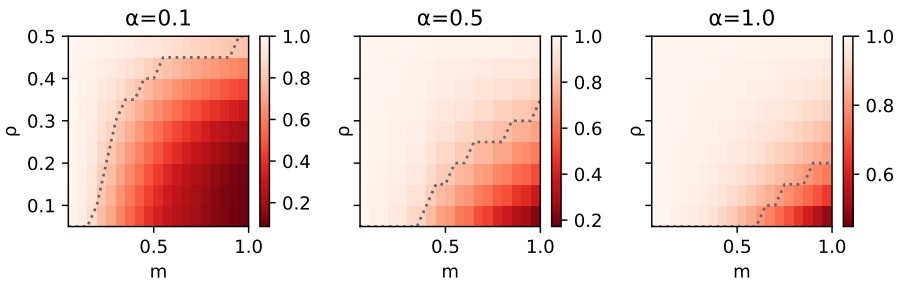

Figure D.2: **Bias with a learnable rule.** We show the accuracy gain as function of the proportion of group $+$ ($\rho$) and the correlation between label and group ($m_+, m_-$). The different figures show how of increasing the dataset size (increasing from left to right) mitigates the bias. The other parameters are: $q_T = 1.0, \Delta_+ = 0.5, \Delta_- = 0.5, b_+ = 0, b_- = 0$.

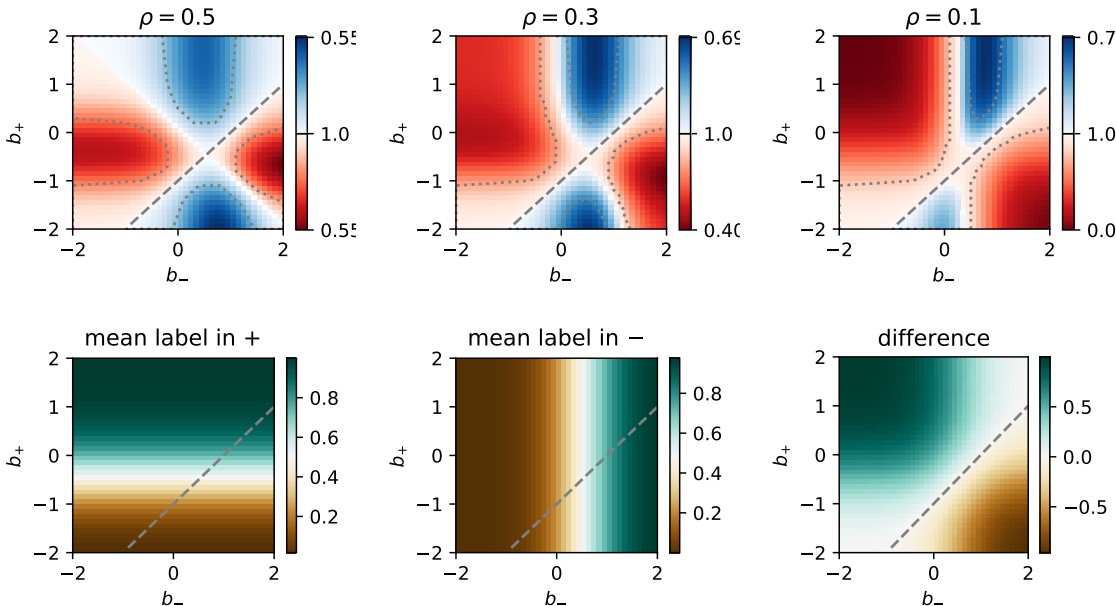

Figure D.3: **Labels within groups and classifier bias.** The *first row* shows the DI as faction of $b_+$ and $b_-$ with $\Delta_+ = \Delta_- = 0.5$, $\alpha = 0.5$, $m_+ = m_- = 0.5$. From left to right, the relative representation $\rho$ moves from equally represented groups to having group $+$ under-represented. The 80% threshold is denoted by the dotted line. The dashed line indicates equal within-group label fraction. The *second row* shows the average labelling in $+$ (left), $-$ (centre), and their difference (right). Notice that these diagrams are independent of $\rho$ and therefor apply to the three settings shown in the first row.

fraction of labels in the T-M model, the bias terms ($b_+$ and $b_-$) are the most relevant since they directly shift the decision boundaries. We consider these two parameters in Fig. D.3 to exemplify this concept.

When the sub-populations are equally represented $\rho = 0.5$, the separations between bias towards $+$ or $-$ is clearly marked by two straight lines. One separation is simply given by the line of equal label fraction, the other is given by the uncertainty of the classifier, receiving contrasting inputs from the two groups. As the relative representation $\rho$ decreases, the classifier accommodates the inputs from the largest group and the separation line is distorted. Finally, observe that the line of equal label fraction (bottom right panel) is not centred in the diagram because $m_+ = m_- \neq 0$.

# E   Mitigation strategies

**Real data.**   In Fig. 6 of the main text, we show the effect of reweighing in the synthetic model. The same analysis can be applied to real data, yielding similar results. In particular, in line with the other validations, we present in Fig. E.1 the result for the CelebA dataset when the splitting is done according to the "Wearing_Lipstick" and the target feature is "Wavy_Hair".

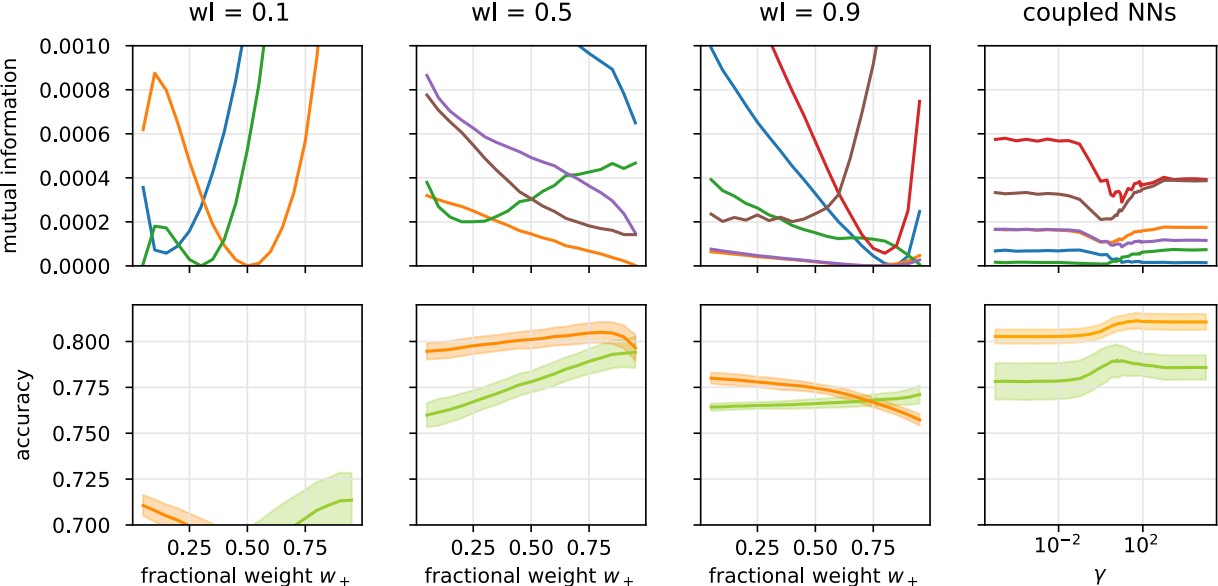

Figure E.1: **Mitigation using re-weighting on real data.** The four panels show the same quantities as in Fig. 6A but applied to the CelebA dataset. Each panels shows in upper figure the mutual information on the fairness metrics – statistical parity, equal opportunities, equal accuracy, equal odds, predicted parity 1, and predicted parity 10– and in the lower figure the accuracy of for subpopulation + and subpopulation −. The first 3 panels show the effect on the reweighing strategy for different values of the label weight (from left to right $w_l = 0.1, 0.5, 0.9$). The last panel shows the performance of the coupled neural network strategy.

Similarly to what observa in the synthetic dataset, the coupled neural network strategy allows for a better performance on all the fairness metrics while retraining a high accuracy for both subpopulations.

**Additional results varying group membership.**   Some strategies require information concerning the group membership of each data point. Depending on the situation, this information may contain errors or it may even be unavailable. Consequently we should take into account the robustness of the mitigation strategies with respect to these errors. Call $\eta$ the fraction of points for which the group was correctly assessed. The phase diagrams in Fig. E.1a show the DI under the reweighing mitigation scheme (controlling the group importance in the loss) and the coupled classifier mitigation. We can clearly observe a greater resilience to the error rates in the case of our strategy. The reweighing strategy appears to have low DI only in extreme cases, where the accuracy on the largest sub-population is greatly deteriorated.

We can understand the larger picture by looking at the different fairness metrics described in the main text, Fig. E.1b, for which the same observations apply. Since $\eta$ is not an actual hyper-parameter, but rather represents an imperfect imputation of the group structure, we consider the maximum for each value of $\eta$. The picture seems quite robust on the side of reweighing (upper group): for every $\eta$ the maximum is achieved for different values of the parameters. Instead, the picture changes for the coupled classifiers (lower group): the method is robust to this perturbation until a critical value (roughly 25% of mismatched inputs), where the minima of the MI become inconsistent and therefore the fairness metrics cannot be optimised all at once.

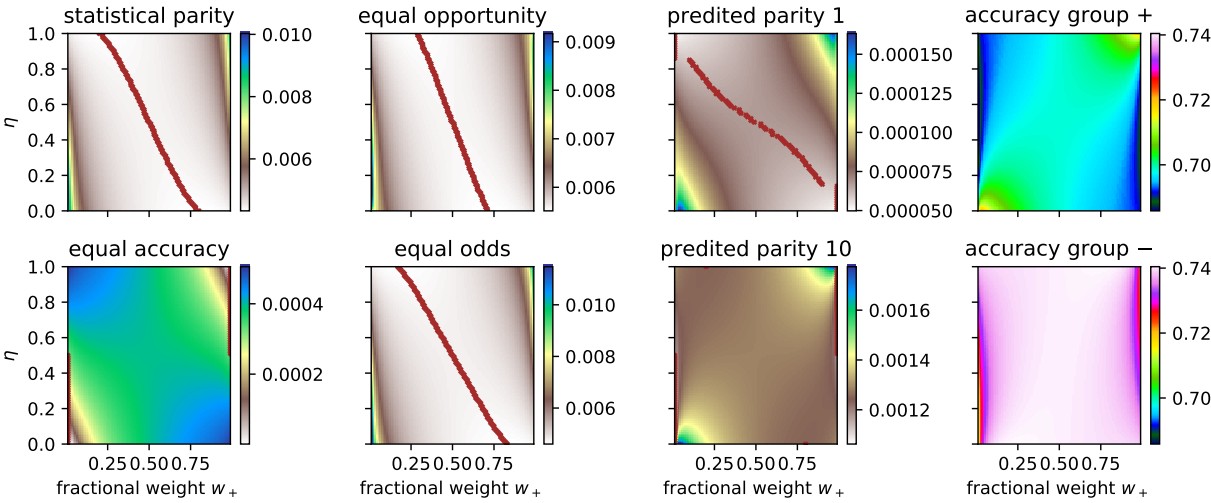

(a) **MI with errors in the group membership under community re-weighting strategy.**

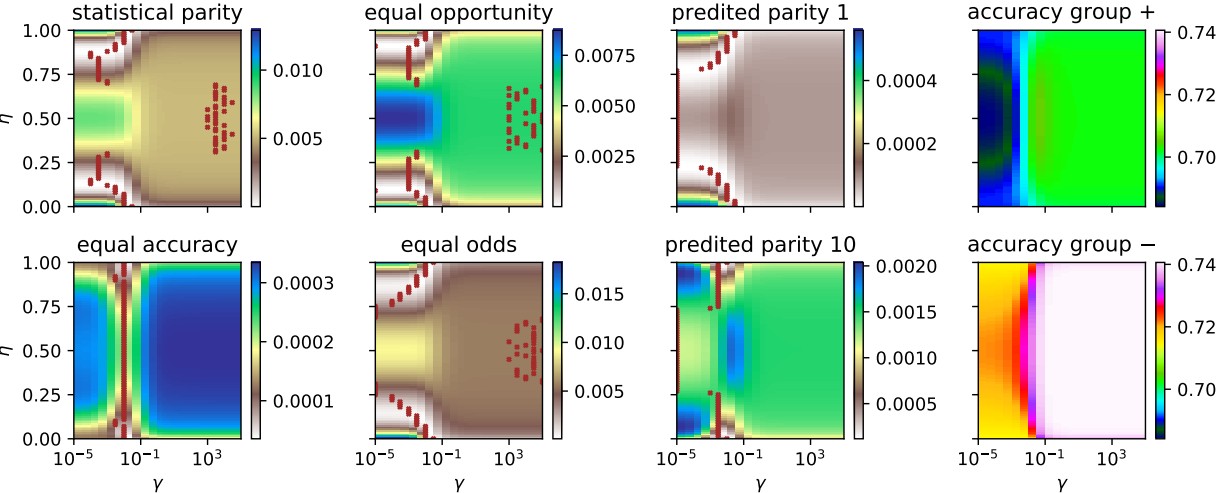

(b) **MI with errors in the group membership under coupled neural networks strategy.**

Figure E.1: **Effect of noise in the attribute of the sub-communities.** In the heatmaps we show in colors how having imperfect information concerning the sub-community membership affect each fairness metrics (six left figures) and the accuracy (right two plots). The vertical axis of the figures represents the probability of mismatch $\eta$, while the horizontal axis refer to the parameter of the strategy ($w_+$ and $\gamma$ resepctively). For every value of the mismatch probability, we denote with red points the minima of the mutual information for each fairness metrics.

