# OpenReview forum: "Bias-inducing geometries: an exactly solvable data model with fairness implications"
_TMLR — Rejected by TMLR_

### Review · Reviewer_D3rk · 2024-08-22

**Summary Of Contributions:**

Authors propose to study the problem of bias for high dimensional unbalanced (and heterogeneous) data. To this aim, the authors introduce a synthetic model, a *Teacher-Mixture* model (based on the Teacher-Student model), which they can analyze exactly. In this model, in the limit of infinite samples and features, the authors can quantify the impact of each parameter of the model (e.g. intragroup variance).

**Audience:**

Yes

**Broader Impact Concerns:**

IMO this paper requires a broader impact statement, which is currently missing.
This paper basically provides potential theoretical guidelines for group partitions in clinical trials, IMO a broader impact statement should be done.

**Claims And Evidence:**

Yes

**Requested Changes:**

- make clearer and lighter figures and captions
- explain in detail and comment on each theoretical result, as well as on the quantities coming into play in the results
- make the link clearer between the experimental results (Figures 2 and 3) and the conclusions
- Overall make the paper much clearer and easy to read

**Strengths And Weaknesses:**

I am far from being an expert in the topic, but I found the paper extremely hard to read, with a lot of typos, uncleared notation, uncommented results, and overpacked Figures. IMO the paper needs to be rewritten (or significantly polished before a second round of review, and then potential publication)

- Figure 1-A: what does the cyan, yellow, and green regions represent? I did not see it in the figure caption. The only mention of this color is the following sentence ¨The shaded areas represent the true regions
of the effectiveness of the tested drug for female/male subjects (cyan/yellow shades¨. Could you elaborate?

- Figure 1-D: the x-label indicates ẗest error¨, while the caption says ¨training performance¨. Is it the training or test error? In addition, I am sure I understood the setup of Figure 1-D: do you train and test on a different with different proportion of classes? I.e. on datasets with different distributions?

- What do you mean by ¨non-rigorous replica method¨?

- Analytical result 1, what is $s$? Analytical result 1 is presented in a very arid way. Could you comment on this result, and develop on why this is interesting? The same comment applies to Analytical result 2.

- I am very sorry but I understood nothing about Figures 2 and 3, would it be possible to give more details? In particular, the conclusion of Section 2 is extremely interesting, but I do not understand the link with Figure 2. Could you elaborate on this?



Minor:
- page 2: there is a line space between theoretical and understanding
- ¨Not surprisingly, when one sub-population a largely predominant on the dataset¨ it seems that a verb is missing in this sentence
- page 6 ¨ Each group needs to achieve equal true positive rate" missing space before the reference
- Typo in analytical result 1 ¨. with¨

---

> ### Author Response · Authors · 2024-10-07
>
> We thank the reviewer for carefully reading our work. Below, please find a list of comments addressing the points raised by the reviewer:
>
> * **Figure 1-A**: cyan, yellow, and green regions: in the description of panel B, we explain how the decision boundaries associated with the colors (and their intersection) are derived. We add a comment in the caption of panel A to refer to that section.
> * **Figure 1-D**: the x-label indicates ẗest error¨, while the caption says ¨training performance¨: We thank the reviewer for spotting this typo.
> * **Setup of Figure 1-D: do you train and test on a different with different proportion of classes?**: We train on an unbalanced mixture of data from the two communities and then test independently on the communities. Therefore, we obtain two separate test errors (the different colors). We added details to clarify in the caption and in the related section in the main text.
> What do you mean by ¨non-rigorous replica method¨?: The replica method is a statistical physics inspired method that is currently regarded as a non-rigorous analytical method. However, in many settings, the results yielded by the replica method were proven correct via different (rigorous) proof methods, and in general it is believed to provide exact results in convex settings like the one analysed in this work. In order to show the validity of the results, we plotted in Fig.1-D simulation and theoretical line and they display a perfect alignment.
> * **Analytical result 1, what is s? … is presented in a very arid way … same for result 2**: ‘s’ is the so-called entropic term of the average free-entropy associated to the learning problem. By imposing an extremisation condition of the free-entropy one obtains a set of simple iterative equations that converge to the typical values of the order parameters (i.e. sufficient descriptors) of the minimizers of the training loss. The equations are reported in the result statement, but we added a clarification on how to interpret the quantities ‘s’ and ‘e’ which are involved in the extremisation. Brief comments on the utility of the results can be found right below the statements.
> * **I understood nothing about Figures 2 and 3 … give more details? … link with Figure 2 … make clearer and lighter figures and captions**: We splitted the figures and caption to make them visually lighter. However, we tried to have self-explanatory captions (therefore verbose) in order to convey the information necessary to read the plots, we hope that in the splitted the format they will appear lighter as well without reducing their content. We revised the text in the captions and main text where we felt was less clear, hoping the reviewer can see throught the explanations of the results.
> * **Comment on the quantities coming into play in the results … make the paper much clearer and easy to read**: Following the reviewers comments we added a table (in the supplementary) with a list of the parameters appearing in the results. Moreover, we added some connections to the paragraphs where their interpretation is presented.
> * **Minors**: We thank the reviewer for his indication, we revised the text to account and correct for them.
> * **This paper requires a broader impact statement … paper basically provides potential theoretical guidelines for group partitions in clinical trials**: We do not provide theoretical guidelines in this work, but rather propose a simple model where some of the potential complications of biased sampling can be analyzed under parametric control. For this reason, we don’t see an immediate impact of this work on any guideline for group partitions in clinical trials.

---

### Review · Reviewer_mHcN · 2024-08-31

**Summary Of Contributions:**

The paper proposed a new theoretical framework, called Teacher-Mixture (TM) model, as a controlled synthetic setting to analyze bias and fairness in neural networks. The TM model involves a data generation model based on Gaussian mixtures with two centers and a linear labeling function. Assuming a single-layer neural network, the paper derived exact analytical characterizations of the optimal network weights in terms of TM model parameters and classification accuracies on different sub-populations. Using these theoretical results, the paper numerically quantified the unfairness (disparate impact) under different scenarios (i.e., different data distributions and labeling functions in the TM model), to understand bias-induction mechanism. It further analyzed the bias-accuracy trade-off of multiple bias mitigation methods under the setting of their TM model.

**Audience:**

Yes

**Claims And Evidence:**

No

**Requested Changes:**

- In my humble opinion, the writing of theoretical part can be improved. Currently, I need to read back and forth frequently to look for or remind myself of the meaning of each symbol. Some notations are confusing or undefined. Assumptions are scattered and not clearly summarized. Therefore, I personally think it is extremely inefficient and difficult to parse the paper, and I view the theoretical *claims and evidence* as unclear. More details can be found below.

- In "Formal definition", please clarify the following:
  - Does $1/d E [\|W_T^{\pm}\|^2] = 1$ mean $1/d E [\|W_T^{+}\|^2] = 1$ and $1/d E [\|W_T^{-}\|^2] = 1$?
  - What is the product in $W_T^+ \cdot W_T^-$
  - In the definition of $y^\mu$, what do $W_T^{\pm}$, $x_{\pm}^\mu$ and $b_T^{\pm}$ represent? The expression gives the impression that $y^\mu$ have two values.
  - $b_s$ in Eq. 1 is undefined. $b$ in the line above Eq. 1 is undefined. It seems both $b$ and $b_s$ represent the bias of student network. If so, please use the same notation and clarify the definition.
  - Below Eq. 1, "where ... the index $c_\mu \in \{+, -\}$ denotes the group membership of data point $\mu$." There is no $c_\mu$ in Eq. 1. Please clarify what this $c_\mu$ refer to.

- In Analytical result 1,
  - "the scalar descriptors ... of the vector $w$ obtained by the empirical risk minimisation of Eq. 1..." Should it be $w$ or $W$? Please clarify.
  - $\delta q$ is undefined.
  - $b$ is not an argument in $e(\Theta; \Delta)$. What does $\partial_b e(\Theta; \Delta)$ represent?
  - "$w$ is the soluiton of ...". What does $w$ refer to here?
  - What does $y$ represent in Eq. 3 and $\ell(y, ...)$?

- In Analytical result 2,
  - What does $\tilde{b}_c$ represent?

- I suggest using fewer symbols or use a table to give a unified definition of all symbols used in the paper.

- Analytical results 1 and 2 can be organized as Theorems, with a clear statement of all the assumptions and notations within, making them understandable as standalone results.

- In Section 2, please clarify details about how DI is calculated. Based on my understanding, DI is calculated using Analytical results 1 and 2. However, a clear expression for DI is missing. Can the authors provide explicit expressions of DI in terms of the parameters considered in the experiments, e.g., $q_T$, $\rho$, $ \Delta_{\pm}$, etc.? Is DI calculated analytically, or is numerical approximation (e.g., for expectations) involved?

- Weird formatting. Several paragraphs are highlighted in purple. Did the authors intend to highlight them?
- Typos, e.g.,
  - "Despite many empirical successes, a large gap remains in the theoretical (new line) understanding of bias-induction mechanisms and how to counteract them. The introduction of a controlled minimal setting, where these phenomena can be characterised exactly could allow for a better theoretical grasp of these nuanced interactions."

**Strengths And Weaknesses:**

**Strengths**:
- Addressed the lack of a controlled theoretical setting for analyzing bias in deep neural networks; allowed exact theoretical analysis of the interactions of bias, data distribution and labeling function; facilitated the understanding of bias-induction mechanisms in a simplified setting.



**Weaknesses**:
- The theoretical results of the paper can be better organized and more clearly presented. Currently many notations are confusing or undefined. Assumptions in the analytical results are not clearly stated. As a result, it is very inefficent to read the paper and difficult to understand the claims and evidence.


- The theoretical analysis and experimental results are focused exclusively on single layer neural networks, which limits the scope of the findings.

---

> ### Author Response · Authors · 2024-10-07
>
> We thank the reviewer for carefully reading our work and helping us improve the clarity of the presentation. Below, please find a list of comments addressing the points raised by the reviewer:
> * **Many notations are confusing or undefined … Assumptions in the analytical results are not clearly stated … inefficient to read the paper and difficult to understand the claims and evidence**: We apologize to the reviewer for the presence of many typos and unclear sections, induced by partially implemented changes in notation and the intention of being coincise. We followed up on the precise comments of the reviewer to correct and unify the notation, introduce the parameters properly, and add references to the paragraphs with the parameter definitions to help the parsing.
> * **Fewer symbols … or a table to give a unified definition of all symbols**: We agree with the reviewer that the setup requires a very large amount of parameter definitions, and the reading is hindered. However, it is not possible to reduce their number without limiting the flexibility of the model. We thus followed the second proposal of the reviewer and added a table in the supplementary recapping all the parameter definitions.
> * **Analytical results 1 and 2 can be organized as Theorems ... clear statements of assumptions and notations**: Unfortunately, the replica method used to derive the results is deemed to be correct but is regarded as a non-rigorous method, therefore our analytical statements cannot be treated as theorems. We will however add clear references to the assumptions and notations used in the presented statements.
> Clarify details about how DI is calculated … explicit expressions of DI in terms of the parameters considered in the experiments … DI calculated analytically … numerical approximation (e.g., for expectations) involved?: The reviewer is correct, the analytical result 2 provides the expression that can be separately used to obtain the numerator and denominator of the DI. The expression needs to be evaluated numerically (Gaussian expectation cannot be taken analytically) and cannot be reduced to a simpler expression of the model parameters.
> * **Several paragraphs are highlighted in purple**: We used the example of the drug testing problem as a tool for introducing the ideas behind the modeling setup of the paper, and we highlighted the associated sections in purple to aid the reading. This type of formatting is often used in different journals, but if it is problematic for TMLR we can remove it.
> * **Typos**: We thank the reviewer for pointing this out, we tried to spot and fix the typos in our paper.
> * **Focused exclusively on single layer neural networks … limits the scope of the findings**: We fully agree with the reviewer on this point, but this limitation is currently unavoidable with the analytical techniques employed in this work. While expanding at least the empirical part to deeper networks is certainly an important avenue of research. Unfortunately, studying the asymptotic performance in deeper network is an open problem except in a few limited cases. Already, the results in the single layer case showed novel and interesting results, nevertheless, we agree with the reviewer’s point and we are leaving this direction for future work. We expanded the “Discussion” section to highlight limitations and possible extensions of the model in the light a of recent theoretical results.

---

### Review · Reviewer_xt6c · 2024-09-23

**Summary Of Contributions:**

The submission seems to propose the Teacher-Mixture (T-M) model, aiming at
exploring biases in machine learning.  The T-M model seems to be an adaptation
of existing frameworks (Gaussian Mixture and Teacher-Student models). The
authors claim to uncover bias mechanisms tied to data geometry, but these
observations seem intuitive and not groundbreaking.  The authors also provide
analytical results and Bias Mitigation Strategies (loss reweighing and coupled
neural networks). However, these strategies seem to be standard approaches in
the fairness literature.

**Audience:**

Yes

**Claims And Evidence:**

No

**Requested Changes:**

1. Improve the presentation (Critical). Define important terms and symbols early
   for smoother understanding. Ensure all figures are self-explanatory. Do not
   assume readers knows all the terminology.
2. Discuss potential issue when the T-M model does not capture real-world
   complexity, such as non-linearity, non-Gaussian features.
3. Show innovations in the proposed models, methods, and strategies.
4. Expand empirical validation.

**Strengths And Weaknesses:**

Strengths:

1. The paper focus on the important topic of biases and fairness in machine
   learning.
2. The paper claims exact analytically solvable exploration of bias-inducing
   factors.
4. The paper tries to isolates different sources of biases.

Weaknesses:

1. The presentation of the needs improvement. For example, which symbol
   represents male and which symbol represents female in Figure 1 A? The paper
   heavily rely on the T-M model, but what is it? The relative representation is
   important and used in Figure 1 on page 3, but it is defined in Remark 1 on
   page 5.
2. The T-M model seems overly simplistic, based on controlled, artificial data
   structures.
3. Some concepts strategies such as loss reweighing, Gaussian mixture
   distributions, and coupled network have been extensively explored in both
   theoretical and empirical settings.
5. Despite some connections to real-world scenarios (e.g., drug testing examples
   or CelebA dataset), the work primarily focuses on an ideal mathematical
   model. In view of this, empirical validation provided by the paper is
   limited, only one real-world data example on the CelebA dataset.

---

> ### Author Response · Authors · 2024-10-07
>
> We thank the reviewer for carefully reading our work. Below we list direct answers to the points raised in this review:
> * **Bias mechanisms tied to data geometry … intuitive and not groundbreaking**: We agree with the reviewer that the link between data geometry and bias induction is intuitive, we are not claiming it to be groundbreaking. In contrast to other problems in machine learning, fairness doesn’t yet have a simple model that can be studied. This is a severe limitation for theory, where we need to know the probability distribution of the data in order to derive precise statements. Indeed, the point of our study is more to show that despite the extreme simplifications entailed in our data model (the referee also thinks the T-M is overly simplistic), a rich bias phenomenology can be induced, similar to that observed on realistic datasets. Moreover, the simplifications allow parametric control over the induced bias, and at the same time make the theoretical analysis feasible.
> * **Loss reweighing and coupled neural networks … standard approaches in the fairness literature … loss reweighing, Gaussian mixture distributions, coupled network have been extensively explored in both theoretical and empirical settings**: We do describe the loss reweighting strategy as a well-known and possibly naive approach for mitigating bias, citing previous work. However, to our knowledge, the coupled networks strategy described in the paper is similar in spirit but not identical to previously considered approaches. Since the referee seems to claim the approach is not novel, could they please provide a reference where such an approach is presented and analysed? We would be happy to comment on similarity and differences. In the paper, we cited the previous work and framed our model as a mixture of previously analyzed models (Gaussian mixture, teacher-student setups), where the co-presence of features from these models induces a richer phenomenology, which was not studied before in such theoretical works.
> * **Which symbol represents male and female? … definition of the relative representation in Figure 1 on page 3 … ensure all figures are self-explanatory**: We are slightly surprised that the referee found the use of the standard gender symbols unclear. We can however add a clarification in the text if this is deemed necessary. We are going to clearly define all the quantities appearing in figure 1 immediately in the caption to improve readability.
> * **Expand empirical evaluation … limited, only one real-world data example on the CelebA dataset**: Indeed, the focus of our paper is not on retracing the phenomenology of the T-M in real datasets, but rather the converse. We believe the phenomenology we find in the T-M was already observed in many study cases in the literature: the various bias effects are not too surprising, what is more surprising is what can be a sufficient cause for their introduction in a simple model like the T-M. For this reason, we only mention experiments on CelebA in the main text, which is a standard dataset in fairness studies. In Appendix C, we instead show some results on the Medical Expenditure Panel Survey (MEPS) dataset, where we observed similar phenomenology. Perhaps, the reviewer can suggest a dataset that could lead to a different phenomenology, or pinpoint bias effects that need more empirical confirmation.
> * **When the T-M model does not capture real-world complexity, such as non-linearity, non-Gaussian features**: We fully agree that the T-M is not sufficient to understand all the subtleties of real-world fairness problems. Given the observation of the reviewer, we expanded the “Discussion” section where these limitations (that are ubiquitous in theoretical analyses) are mentioned referencing to recent results that could be used to address some of them.

---

### Decision · Action_Editor_zQAq · 2024-11-19

**Recommendation:** Reject

**Comment:**

To put it briefly, this paper analyzes the disparate impact of neural networks by deriving its analytical form with the replica method; to this end, we can investigate how each system parameter contributes to the potential bias. Despite that some reviewers have concerns about its over simplicity, I do believe the analytical model studied in this paper is an important step to pave a road to further understanding of ML bias.

During the review phase, all reviewers unanimously raised concerns about the clarity of the paper. This is partly due to several typos and lack of symbol definitions---which the authors have adequately addressed in the revision. Besides, the interpretation of the analytical results was not straightforward in the submitted version partly because the replica analysis might not be common in the ML community yet. Nonetheless, the authors have greatly improved the presentation in the revision.

Since we still need another round to verify the paper's claim, I would highly recommend the authors resubmit.

**Audience:**

This paper deals with the disparate impact of neural networks under the analytical model, which is certainly relevant to a broad audience of TMLR.

**Claims And Evidence:**

In the initial version, the results of the replica method and phase diagrams were challenging to interpret, which hinders readers from understanding the claims of this paper. I gratefully acknowledge the authors' efforts to improve these presentations after the discussions with the reviewers.

**Resubmission Of Major Revision:**

The authors may consider submitting a major revision at a later time.